# Allosteric mechanism of signal transduction in the two-component system histidine kinase PhoQ

Bruk Mensa[1,2,3]*, Nicholas F Polizzi[1], Kathleen S Molnar[4], Andrew M Natale[1,2,5], Thomas Lemmin[6], William F DeGrado[1,2]*

[1]Department of Pharmaceutical Chemistry, University of California, San Francisco, San Francisco, United States; [2]Cardiovascular Research Institute, University of California, San Francisco, San Francisco, United States; [3]Chemistry and Chemical Biology PhD program, University of California, San Francisco, San Francisco, United States; [4]Codexis Inc., Redwood City, United States; [5]Biophysics PhD program, University of California, San Francisco, San Francisco, United States; [6]Euler Institute, Università della Svizzera Italiana, Lugano, Switzerland

**Abstract** Transmembrane signaling proteins couple extracytosolic sensors to cytosolic effectors. Here, we examine how binding of $Mg^{2+}$ to the sensor domain of an *E. coli* two component histidine kinase (HK), PhoQ, modulates its cytoplasmic kinase domain. We use cysteine-crosslinking and reporter-gene assays to simultaneously and independently probe the signaling state of PhoQ's sensor and autokinase domains in a set of over 30 mutants. Strikingly, conservative single-site mutations distant from the sensor or catalytic site strongly influence PhoQ's ligand-sensitivity as well as the magnitude and direction of the signal. Data from 35 mutants are explained by a semi-empirical three-domain model in which the sensor, intervening HAMP, and catalytic domains can adopt kinase-promoting or inhibiting conformations that are in allosteric communication. The catalytic and sensor domains intrinsically favor a constitutively 'kinase-on' conformation, while the HAMP domain favors the 'off' state; when coupled, they create a bistable system responsive to physiological concentrations of $Mg^{2+}$. Mutations alter signaling by locally modulating domain intrinsic equilibrium constants and interdomain couplings. Our model suggests signals transmit via interdomain allostery rather than propagation of a single concerted conformational change, explaining the diversity of signaling structural transitions observed in individual HK domains.

## Editor's evaluation

This paper examines the mechanism of signal transmission through the histidine kinase PhoQ. The paper nicely describes a model of signaling by allosteric coupling between individual domains rather than by a concerted conformational change and provides substantial experimental evidence for the model from characterization of over 30 mutational substitutions. The allosteric coupling model provides a way to understand many diverse observations about signaling by two-component receptors and has the potential to be relevant to conformational signaling by many other transmembrane receptors.

## Introduction

Two-component system sensor Histidine Kinases (HKs) are conserved signaling modules in bacteria responsible for sensing a myriad of environmental stimuli and orchestrating transcriptional responses

*For correspondence:
bruk.mensa@ucsf.edu (BM);
Bill.DeGrado@ucsf.edu (WFDeG)

**Figure 1.** Modular architecture of histidine kinases. Various protein folds and numbers of signal transduction domains are found inserted between sensor (blue) and autokinase (purple). Structurally elucidated examples include simple coiled-coils (NsaS), HAMP (AF1503), PAS (VicK), GAF (Nlh2), Tandem HAMP (Aer2), and HAMP/PAS domain (VicK). PDB codes are provided in figure, except for NsaS (NMR structure).

along with their cognate transcription factors (Response Regulators, RR) (*Stock et al., 2000*; *Groisman and Mouslim, 2006*). These sensors are generally implicated in environment sensing and are involved in multi-drug resistance (*Nishino et al., 2005*; *Hirakawa et al., 2003*; *Nizet, 2006*) and as master regulators of virulence programing in pathogenic bacteria (*Thomas and Cook, 2020*; *Delauné et al., 2012*). HKs are constitutive homodimers, which transmit signals through a series of intermediary domains to a cytoplasmic catalytic domain. While the lack of a full-length HK structure has hampered our understanding of the mechanism of signal transduction in these proteins, cytoplasmic domain structures have shed light particularly on the enzymatic core of this class of kinases. Several crystallographic snapshots of the autokinase domains of multiple HKs in various conformations (*Ferris et al., 2014*; *Rivera-Cancel et al., 2014*; *Wang et al., 2013*; *Mechaly et al., 2014*; *Mechaly et al., 2017*; *Casino et al., 2009*; *Albanesi et al., 2009*), particularly CpxA, DesK, and VicK, have shown distinct conformations involved in autophosphorylation, phosphotransfer, and dephosphorylation that may be conserved across this family. While these structures offer a conserved view of the catalytic cycle of the cytosolic autokinase domain (*Jacob-Dubuisson et al., 2018*), the question of how these proteins couple a sensory event on the other side of the membrane, and many nanometers away to the modulation of the activity of this domain remains unanswered.

This question is especially perplexing in light of the various modular architectures of HKs, involving the insertion of one or more signal transduction domains between sensors and the conserved autokinase domain. It is abundantly clear that the same conserved autokinase domain that defines this protein class can be regulated by a myriad of structural inputs, ranging from short alpha-helical dimeric coiled coils, to well-folded tertiary folds such as HAMP, PAS, and GAF domains (*Krell et al., 2010*; *Bhate et al., 2015*; *Figure 1*). Moreover, it is clear from the representation of these folds in diverse protein classes that these domains evolved independently of HKs and were incorporated pervasively into functioning HK architectures. Therefore, they are likely to serve a generalizable function that is robust to evolutionary selection and allow for the construction of physiologically relevant sensors optimally positioned to respond to environmental changes. While some intervening transduction domains have clearly annotated functions, such as the binding of intracellular ligands which are integrated into the sensory function of the HK, the requirement for other signal transduction domains remains obscure.

In this work, we evaluate the coupling of sensor and autokinase domains in a model Gram-negative HK, PhoQ (*Miller et al., 1989*), in which these domains are separated by intervening transmembrane

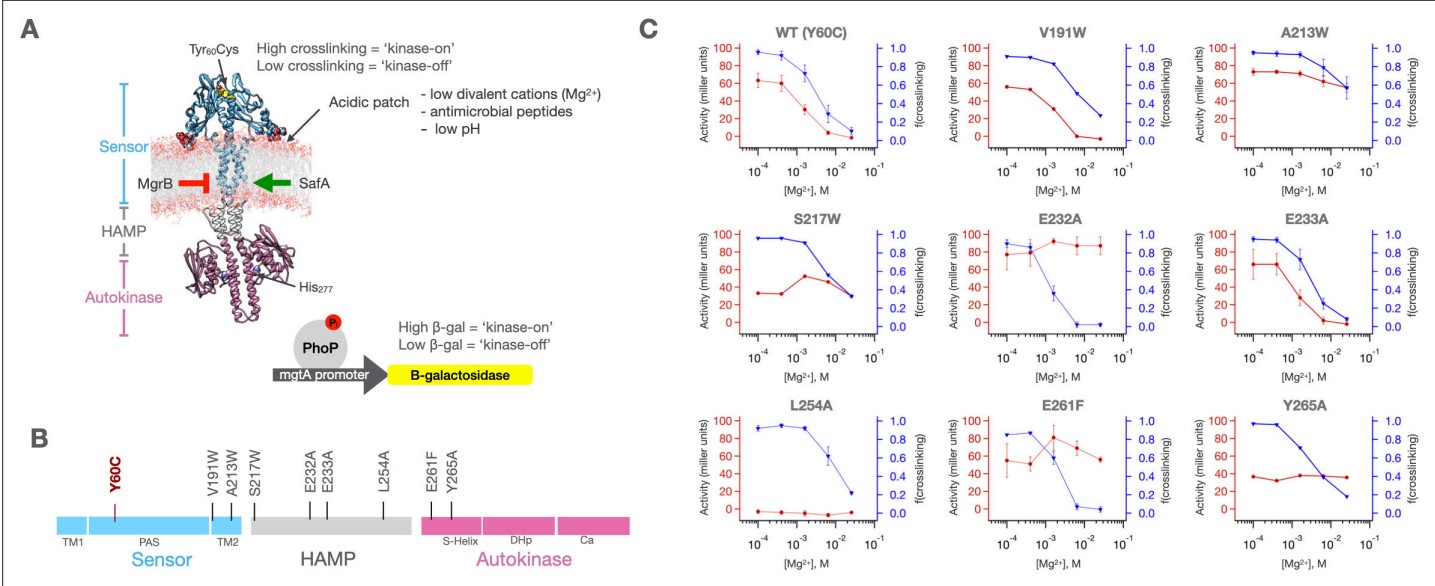

**Figure 2.** PhoQ single mutants exhibit a range of behaviors. (**A**) Molecular Dynamics model of dimeric PhoQ in which the sensor (res. 1–219, blue), HAMP (res. 220–260, grey) and autokinase domains (res 261–494, purple) are annotated. The sensor contains a Y60C mutation (spheres) which shows signal state dependent crosslinking. The autokinase contains the conserved catalytic His277, which upon phosphorylation transfers a phosphoryl group to the response regulator PhoP, which then modulates a *mgtA* promoter-driven β-galactosidase reporter. Stimuli and regulatory proteins that modulate PhoQ activity are shown. (**B**) Linear topology diagram of PhoQ. The sensor, HAMP and Autokinase domains are highlighted in blue, gray, and purple, respectively. The locations of mutations in panel (**C**) are shown. (**C**) Fraction of sensor crosslinking (blue) and autokinase activity (red) determined for 'wild type' (Y60C) PhoQ, as well as eight mutants along the signal transduction pathway (n = 9 for WT, n = 2 for A213W, E232A, E233A, L254A, and E261F, n = 1 for V191W, S217W, and Y265A). The sensor state and autokinase activity do not show identical ligand-dependent behavior as would be predicted by a concerted signaling mechanism. Error bars correspond to± SD, where applicable.

The online version of this article includes the following source data and figure supplement(s) for figure 2:

**Source data 1.** [Mg2+] dependent activity and sensor crosslinking of mutants in *Figure 2C*.

**Figure supplement 1.** Representative PhoQ crosslinking western blot and quantification of WT (Y60C) and Y60C/L224A mutant.

**Figure supplement 1—source data 1.** Western blot image for PhoQ sensor crosslinking of WT (Y60C) and L224A mutant.

and HAMP signal transduction domains. The PhoQP two-component system is composed of a canonical transmembrane sensor HK, PhoQ, that senses the presence of divalent cations (*García Véscovi et al., 1996*; *Soncini et al., 1996*) and polycationic species such as antimicrobial peptides (*Bader et al., 2005*; *Hancock and McPhee, 2005*), and a cognate response regulator, PhoP (*Miller et al., 1989*; *Groisman et al., 1989*), which transcriptionally controls regulons related to cation transport and outer-membrane remodeling (*Fields et al., 1989*; *Behlau and Miller, 1993*; *Belden and Miller, 1994*; *Gunn and Miller, 1996*; *Guo et al., 1997*; *Bearson et al., 1998*; *Guo et al., 1998*; *Adams et al., 2001*; *Bader et al., 2003*; *Dalebroux et al., 2014*). The kinase activity of PhoQ is repressed by divalent cation binding, whereas it is enhanced by the presence of antimicrobial peptides. PhoQ is additionally implicated in low pH sensing (*Prost et al., 2007*) via an interaction with the membrane protein UgtL (*Choi and Groisman, 2017*), and has more recently been suggested to respond to changes in osmolarity (*Yuan et al., 2017*). With respect to its most well characterized function, that is, the sensing of divalent cations such as Mg²⁺, it is hypothesized that in the absence of such cations, the electrostatic repulsion between an acidic patch in the sensor domain and the negatively charged bacterial inner membrane enforces the 'kinase-on' conformation of the sensor and results in high-kinase/ low-phosphatase activity in the autokinase domain. In the presence of divalent cations, the electrostatic interaction between the sensor and inner-membrane are bridged resulting in a different 'kinase-off' sensor conformation that corresponds to low-kinase/ high-phosphatase autokinase function (*Cho et al., 2006*; *Chamnongpol et al., 2003*).

To probe the coupling between the sensor and autokinase domains, we established two assays, which allow simultaneous measurement of the conformational states of the sensor and autokinase domains (*Figure 2A*). Like most HKs, PhoQ is a constitutive parallel homodimer, in which the individual

domains interact along a series of coaxial helical bundles. Previously, we observed that a Tyr60 to Cys variant forms interchain disulfides between the two monomers only in the absence of $Mg^{2+}$ where the protein is in the 'kinase-on' state (*Molnar et al., 2014*). Thus, the fraction of the sensor in the 'kinase-on' versus 'kinase-off' state can be readily quantified based on the amount of dimer versus monomer seen in a western blot. Importantly, the Y60C substitution is minimally perturbing, as the $[Mg^{2+}]$-dependent signaling curve for this mutant is nearly identical to wild-type PhoQ with respect to the midpoint of the transition and activity of the basal and activated states. Also, the redox environment of the periplasm of *E. coli* is buffered such that disulfide formation is reversible and hence a good readout of the conformational state of the sensor (*Kadokura and Beckwith, 2010*). To quantify the activity of the auto-kinase domain, we use a well-established beta-galactosidase gene-reporter assay that employs the PhoQ/PhoP-controlled promoter of the $Mg^{2+}$ transporter MgtA. Although this assay is indirect, there is a reasonable correlation between promoter activity and PhoP phosphorylation (*Miyashiro and Goulian, 2007*). We note that similar assays, pairing disulfide crosslinking efficiencies to phenotypic output, have been extensively used by Falke, Haselbauer et al. (*Falke and Hazelbauer, 2001*; *Swain and Falke, 2007*) to probe signal transduction in chemosensors that are related to HKs.

Using this approach, we evaluate the extent to which the sensor's conformational state couples to and dictates the conformational activity of the autokinase domain for a set of over 30 mutations, representing substitutions throughout the signal transduction pathway from the sensor to the autokinase domain. We show how these mutations can modulate the three basic characteristics of a PhoQ signaling response which need to fit the biological role of the HK (1) signal strength at limiting high $[Mg^{2+}]$, (2) signal strength at the limiting low $[Mg^{2+}]$, and (3) the midpoint of the $[Mg^{2+}]$ dependent transition – over the physiologically relevant concentration ranges that *E. coli* encounters (0.1–10 mM). We further evaluate the intrinsic signaling equilibria of the sensor and autokinase domains by disrupting the allosteric coupling between them using poly-glycine insertions in the signal transduction pathway and show that both domains are highly biased to the 'kinase-on' state when uncoupled from each other. The intervening HAMP domain serves as a negative allosteric modulator of both these domains and balances the stability of the 'kinase-on' and 'kinase-off' states so that they can become responsive to physiological concentrations of $Mg^{2+}$. With these concepts in mind, we establish, fit and evaluate a semi-empirical 3-domain allosteric coupling model to account for the sensor-autokinase coupling and high/low asymptote and midpoint of transition behaviors of 35 distinct point-mutations and poly-glycine insertions, and highlight the advantages of inserted signal transduction domains in robustly modulating the signaling behavior of HKs.

## Results
### Single-point mutants along the signal transduction pathway generate a range of sensor and autokinase behavior

We simultaneously measured the sensor cross-linking and autokinase activities of 'wild type' (Y60C) PhoQ and a total of 35 point mutants and sequence insertions at five different concentrations of $Mg^{2+}$ to evaluate the signaling-state correlation of these two domains. Our goal was to investigate the mechanism of signal transduction from the sensor to the kinase. To simplify the interpretation of results we maintained wild-type sequences of the $Mg^{2+}$-binding site and catalytic domain, and mutated at multiple points along the signal transduction pathway. Ala and Phe substitutions were evaluated at sites expected to be on the interior of the protein; these mutants were expected to alter the relative energetics of the kinase- versus phosphatase-promoting states by altering core packing geometry (*Airola et al., 2013*; *Ferris et al., 2011*; *Hulko et al., 2006*; *Bi et al., 2018*). We also examined the effects of Trp substitutions in the TM helix at positions expected to map to the headgroup region of the bilayer, as similar substitutions often induce changes in signaling (*Draheim et al., 2005*; *Lehning et al., 2017*; *Inda et al., 2016*; *Monzel and Unden, 2015*; *Yusuf and Draheim, 2015*; *Yusuf et al., 2018*; *Adase et al., 2013*). We also included several mutations, particularly in the C-terminal half of the HAMP domain, that show altered autokinase activity as compared to WT PhoQ.

Illustrative data in *Figure 2C* show it is possible to generate several combinations of ligand-dependent sensor and autokinase behavior. WT (Y60C) PhoQ had a correlated ligand-dependent response, with the high cross-linking state of the sensor corresponding to high autokinase activity at low $[Mg^{2+}]$, and the low cross-linking sensor state corresponding to low autokinase activity at

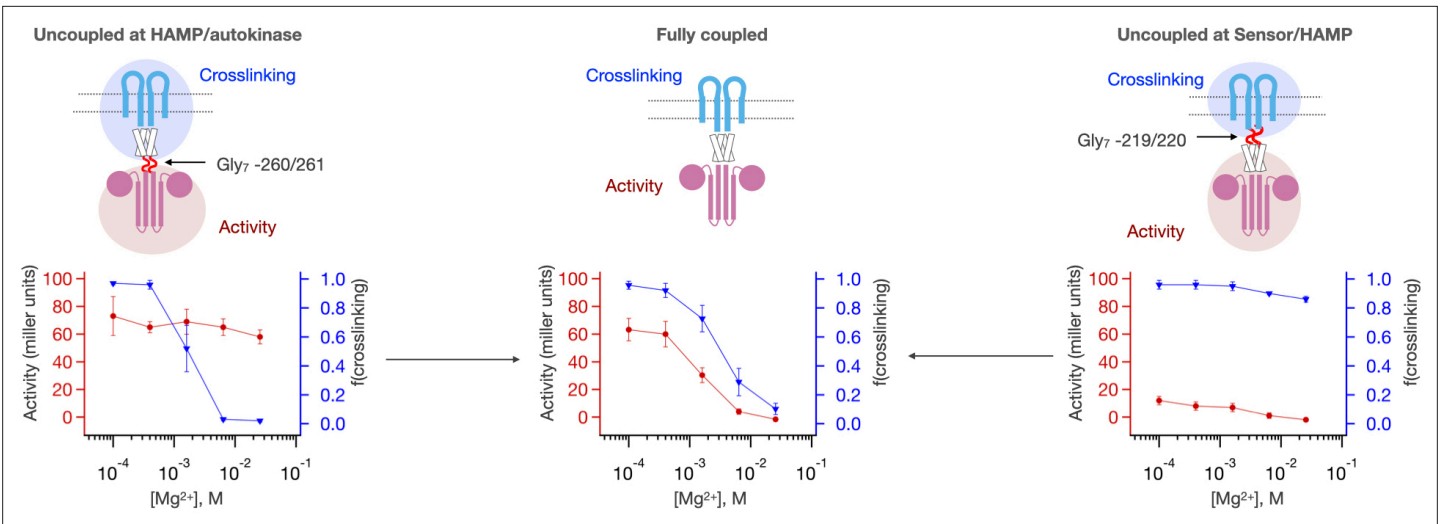

**Figure 3.** Intrinsic activities of the PhoQ sensor and autokinase domains are altered by coupling to HAMP. Gly$_7$ insertions are introduced either between the HAMP domain and the autokinase (Gly$_7$ - 260/261, left, n = 3) or between the Sensor and HAMP domain (Gly$_7$ –219/220, right, n = 2) to disrupt allosteric coupling between sensor and autokinase. Both the sensor and autokinase by themselves show high 'kinase-on' propensity (red trace, left; blue trace, right). The HAMP domain potentiates the 'kinase-off' state, resulting in a more [Mg$^{2+}$] responsive sensor (blue trace, left), or a lower basal activity autokinase (red trace, right). The fully coupled protein shows correlated sensor/autokinase activity(red and blue traces, middle, n = 9).

The online version of this article includes the following source data for figure 3:

**Source data 1.** [Mg2+]-dependent activity and sensor crosslinking of PhoQ Gly7 insertions.

---

high [Mg$^{2+}$] (*Figure 2C*). Some mutants however showed low levels of kinase-activity at low [Mg$^{2+}$] even though the sensor remains in a high-crosslinking 'kinase-on' state (e.g. L254A). Similarly, some mutants retained high kinase-activity in the autokinase despite the fact that crosslinking in the sensor showed WT-like crosslinking in a [Mg$^{2+}$]-dependent manner (e.g. E232A, E261F). Finally, some mutants produced higher levels of kinase activity at low-Mg$^{2+}$ than WT PhoQ (e.g. E232A is more active than WT PhoQ at low [Mg$^{2+}$]). Therefore, mutations along the signal transduction pathway have profound effects in altering or uncoupling sensor-autokinase correlation.

## The effect of decoupling the HAMP domain from the catalytic and sensor domains

We examined the intrinsic activities of the sensor and autokinase domains when decoupled from the HAMP domain, by inserting a stretch of seven helix-disrupting glycines (Gly$_7$) to interrupt the helical connections that are required for coupling between PhoQ's domains. Gly$_7$ insertions were introduced just before the HAMP domain (Gly$_7$ –219/220) as the TM helix exits the membrane and just after the HAMP signal transduction domain within a short helical connection to the autokinase domain (Gly$_7$ –260/261).

As expected, both insertions decoupled Mg$^{2+}$ binding from kinase activity (*Figure 3*). However, they had markedly different effects on the sensor and catalytic domains when these activities are evaluated individually. When the insertion occured between the HAMP and sensor domains, the sensor was highly activated, and remained in the high-crosslinking state, even at concentrations of Mg$^{2+}$ sufficient to switch WT PhoQ to the 'kinase-off' state (*Figure 3*, right). On the other hand, if the HAMP domain remained coupled to the sensor, as in WT or the variant with Gly$_7$ insertion between the HAMP and the catalytic domain (Gly$_7$ –259/260), the sensor behaved normally, being efficiently crosslinked in a [Mg$^{2+}$]-dependent manner (*Figure 3*, left). Thus, the HAMP domain would appear to favor the 'kinase-off' state, serving to reset the energetics of the otherwise highly stable 'kinase-on' state of the sensor.

The HAMP domain had a similar influence on the catalytic domain. When the native connection between the HAMP and the autokinase was disrupted by Gly$_7$ insertion, the autokinase was highly activated (*Figure 3*, right). By contrast, when the connection between the HAMP and catalytic domains was retained as in Gly$_7$ –219/220, the kinase activity was strongly downregulated (*Figure 3*, left). Thus,

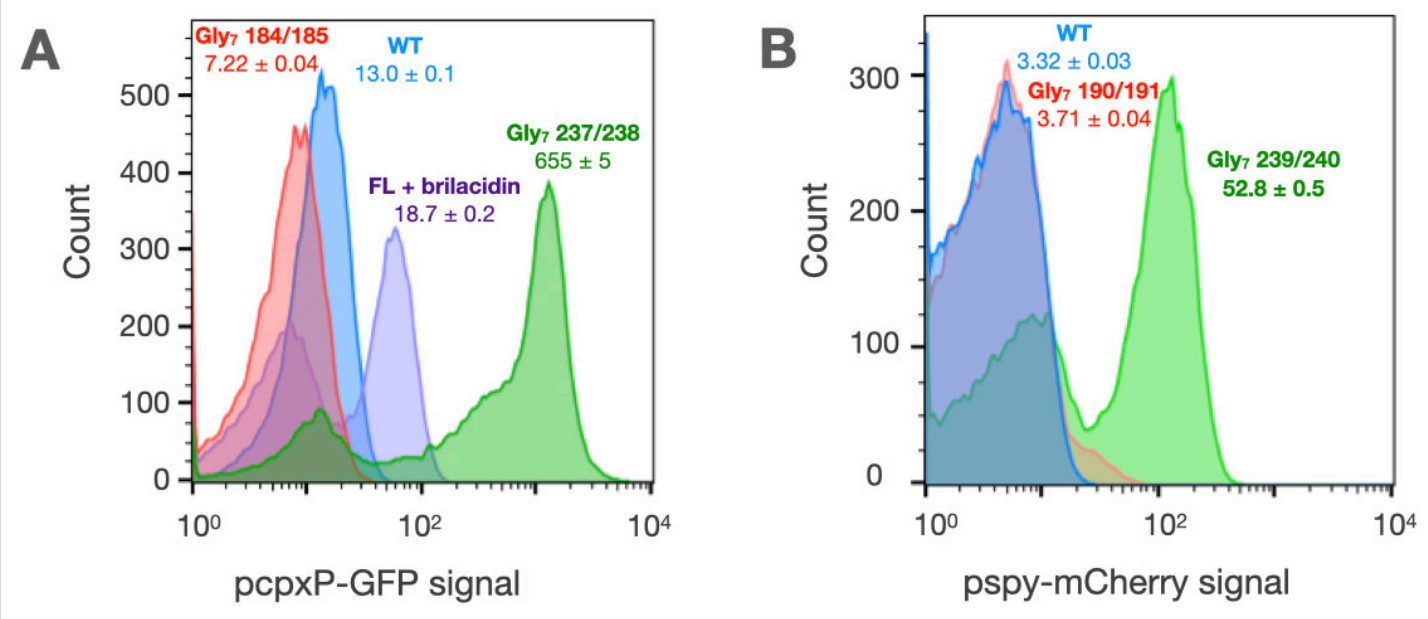

**Figure 4.** Glycine disconnections in CpxA and BaeS. (**A**) The activity of CpxA constructs is measured in AFS51 strain (Δ*cpxA*) using a p*cpxP*::GFP reporter. The activity of WT CpxA (blue) is responsive to the antimicrobial mimetic, brilacidin (*Scott et al., 2008*; *Mensa et al., 2011*) (purple). The autokinase domain of CpxA when uncoupled (Gly7 237/238) shows very high kinase activity (green), which is repressed to basal levels by the addition of the HAMP domain alone (Gly7 184/185, red). (**B**) The activity of BaeS constructs is measured in a Δ*baeS* Δ*cpxA* strain using a p*spy*::mCherry reporter. The autokinase domain of BaeS when uncoupled shows high kinase activity (Gly7 239/240, green) relative to WT (blue), which is repressed by the addition of the HAMP domain alone (Gly7 190/191, red). Median reporter fluorescence values ± STE (n = 20,000) are reported below labels for single experiment.

The online version of this article includes the following source data for figure 4:

**Source data 1.** Raw flow cytometry data for Gly7 insertions in CpxA and BaeS.

the resulting coupling provides an energetic balance so the system can respond to $Mg^{2+}$ over the physiological range.

## The HAMP domain is negatively coupled to the autokinase domains of CpxA and BaeS

Given the profound effect of the HAMP domain on the intrinsic activities of the PhoQ sensor and autokinase domains, we examined if HAMP domains have similar effects in closely related but functionally distinct HKs with the same arrangement of signaling domains as in PhoQ (TM1, PAS sensor, TM2, a single cytosolic HAMP, and the autokinase domains, *Figure 2A and B*). We constructed Gly7 insertions in two closely related *E. coli* HKs, CpxA and BaeS, that have very similar architectures to PhoQ. The HK CpxA responds to periplasmic protein misfolding stress via an accessory protein, CpxP, and upregulates genes to mitigate this stress (*Keller et al., 2015*; *Batchelor et al., 2005*; *Danese and Silhavy, 1997*). It is similar to PhoQ in that the free HK is kinase-active, and is turned off by the binding of the periplasmic CpxP protein (*Zhou et al., 2011*). BaeS is a closely related HK, which has significant overlap with CpxA, both in the inducing stimuli as well as the genes regulated (*Leblanc et al., 2011*). We evaluated the activity of these kinases using previously validated fluorescent gene-reporters p*cpx*-*P*::GFP for CpxA activity (*Clark et al., 2021*), p*spy*::mCherry for BaeS activity *Mensa et al., 2011* in a double CpxA/BaeS knockout strain.

When Gly7 was inserted immediately upstream of the autokinase domain, we observed a high basal activity for both kinases similar to PhoQ (*Figure 4*). However, when the Gly7 motif was placed upstream of the HAMP domain thereby allowing to couple to the autokinase, this high basal activity was potently repressed, again similar to PhoQ. This finding indicates that the HAMP domain strongly coupling to and altering the intrinsic activities of adjacent domains may be a generalizable principle, although it might not serve as a negative element in all cases.

## Fully cooperative and two-domain allosteric models are unable to explain the gamut of activities of mutants

In the following sections, we consider thermodynamic two-state allosteric signaling models of increasing complexity to understand the coupling of the sensor to the autokinase of WT PhoQ and our set of point-mutants. In these models, we assume that $Mg^{2+}$ binds to single sites in the sensor domains. It is possible that binding between the sites is cooperative or that more than one $Mg^{2+}$ ions are bound per domain. However, given the fact that the transcriptional assay is an indirect readout of the 'kinase-on' state, and as such is not necessarily perfectly linear with respect to the fraction of activation (*Miyashiro and Goulian, 2007*), we are unable to differentiate between models that differ subtly in their dose-response curves. However, our data (see below) are able to rule out highly cooperative models in which many binding sites must be occupied with high cooperativity as this would result in a much sharper dose-response curve (*Gestwicki et al., 2000*). We also assume that $Mg^{2+}$ can bind to both 'sensor-off' and 'sensor-on' states, albeit with higher affinity to the 'sensor-off' state, as high $[Mg^{2+}]$ inhibits PhoQ kinase activity. Indeed, for the sensor of PhoQ, there is no reason to preclude ligand binding in either sensor state, since the same negatively charged surfaces are present in both states and can conceivably still bind $Mg^{2+}$, albeit at a much lower affinity due to the lack of bridging interactions (*Waldburger and Sauer, 1996*). Therefore, our model allows for independent $Mg^{2+}$ binding per monomer subunit with built-in stoichiometry factor of 2 in the observed Kds ($K_{dOFF}$, $K_{dON}$), but does not consider similar hybrid activation states in the autokinase.

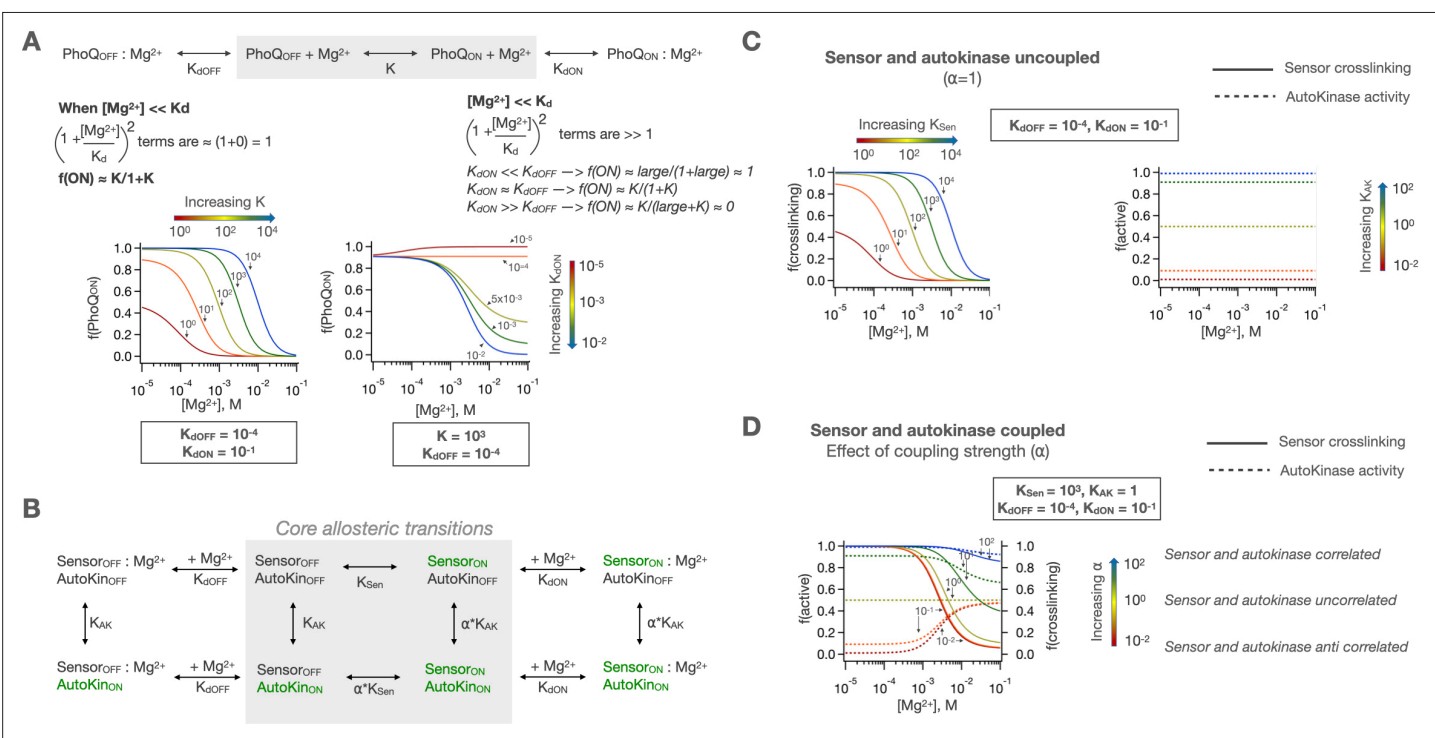

**Figure 5.** Concerted and two-domain allosteric models for PhoQ signaling. (**A**) In a concerted model for signaling, PhoQ has an intrinsic on-off equilibrium (constant = K) which is modulated by $Mg^{2+}$ binding to either states with corresponding $K_d$s. This allows for modulation of both low and high activity asymptotes and the midpoint of transition but requires perfect correlation between sensor and autokinase signaling states. Equations for calculating population fractions are shown in Methods (*Equation 1*). (**B**) The sensor and autokinase domains of PhoQ are allowed to sample both 'kinase-on' and 'kinase-off' states with equilibrium constants $K_{Sen}$ and $K_{AK}$ when the other domain is in the 'kinase-off' state. When the other domain is in the 'kinase-on' state, the equilibria are scaled by the coupling constant, $\alpha$. This allows for semi-independent fractions of sensor and autokinase in the 'kinase-on' state, which are computed as shown in Methods (*Equation 2*). (**C**) In the uncoupled case ($\alpha = 1$), $K_{Sen}$ modulates the sensor identically to the previously described concerted signaling mechanism, while $K_{AK}$ sets the basal autokinase activity. (**D**) The coupling of these domains with $\alpha \neq$ one results in $[Mg^{2+}]$ dependent activity that is either correlated ($\alpha > 1$) or anticorrelated ($\alpha < 1$). As $\alpha$ gets larger, the two domains act more as one concerted protein.

The online version of this article includes the following figure supplement(s) for figure 5:

**Figure supplement 1.** Effects of $K_{Sen}$ and $K_{AK}$ on two-domain signaling.

The simplest model for signaling in HKs is one in which the entire HK exists as one concerted domain in a two-state equilibrium of 'kinase-on' and 'kinase-off' states (equilibrium constant = K) which is then modulated by ligand binding (*Figure 5A*). In such a model, PhoQ will be pushed to a fully ligand bound, 'kinase-off' state at high enough $[Mg^{2+}]$ ($[Mg^{2+}] >> K_d$). At low $[Mg^{2+}]$, the activity of PhoQ is determined by the partition of PhoQ into the low and high affinity $Mg^{2+}$ bound states according to the ratio of the two dissociation constants, $K_{dOFF}$ versus $K_{dON}$. The midpoint of transition is dictated by the relative magnitudes of K, which reflects PhoQ's preference for the 'kinase-on' vs. 'kinase-off' state, as well as the two $K_d$s, as shown in *Figure 5A*. However, this model cannot explain why several mutants of PhoQ do not show a 1-to-1 correlation between their sensor and autokinase signaling states (*Figure 2C*). Moreover, some mutants also have higher autokinase activity than WT PhoQ at low $[Mg^{2+}]$, demonstrating that even at the low-$[Mg^{2+}]$ conditions in which the sensor is fully in the crosslinked 'kinase-on' state, there remains a significant fraction of the WT autokinase that remains in the 'kinase-off' state. Therefore, this fully concerted signaling model is insufficient to describe the full range of activities of PhoQ variants.

The next model we considered is one in which two domains, sensor and autokinase exhibit two-state equilibria and communicate allosterically. A ligand-dependent sensor can be allosterically coupled to an autokinase domain with a tunable coupling strength to allow for the desired degree of communication between the sensor and the autokinase. In such a scheme, the sensor would be a ligand-binding domain with all the properties previously described for a fully concerted HK. The autokinase on its own would have a constant activity level based on its own intrinsic 'kinase-off' to 'kinase-on' equilibrium. The sensor is then connected to the autokinase in a manner that biases the intrinsic autokinase equilibrium differently depending on which signaling state the sensor is in. A ligand-dependent allosterically modulated HK results from such a coupling, so long as sensor 'kinase-on' and 'kinase-off' states of the sensor alter the autokinase equilibrium differently (*Figure 5B and C*).

To reduce the number of parameters needed to describe such a model, we can define the intrinsic equilibria of the sensor and autokinase when they are connected to a reference state (e.g. 'kinase-off') with equilibrium constants as shown in *Figure 5B*. $K_{Sen}$ is the 'intrinsic' equilibrium of the sensor domain when connected to an autokinase in the 'kinase-off' state, and $K_{AK}$ is the 'intrinsic' equilibrium of the autokinase domain when connected to the sensor in the 'kinase-off' state. When coupled to the 'kinase-on' state of either domain, $K_{Sen}$ and $K_{AK}$ are scaled by a new factor, α. *Figure 5C and D* illustrate the effect of α on the $Mg^{2+}$ dose-response curves. When α = 1, the two domains are fully uncoupled, and the binding of $Mg^{2+}$ to the sensor is unable to affect the autokinase domain (*Figure 5C*). A value of α > 1 means that when either of the domains switches to the 'kinase-on' state, the other domain's propensity to switch 'kinase-on' state is also enhanced by that factor, creating a correlated ligand-mediated transition between sensor and autokinase (*Figure 5D*). If 0< α < 1, then a transition to 'kinase-on' state is actually easier when the other domain is in the 'kinase-off' state, creating an anticorrelated ligand dependent behavior. When the absolute value of the log of α becomes very large (i.e. when α is either >>1 or approaching zero), the two domains are highly coupled (*Figure 5D*) and the system behaves as in the fully concerted 2-state models in *Figure 5A*. Therefore, α is the coupling strength between the 'kinase-on' states relative to the coupling strength between the 'kinase-off' states already accounted for in $K_{Sen}$ and $K_{AK}$.

Coupling provides a robust mechanism for setting both the upper and lower activity asymptotes of the WT sensor kinase. At high enough $[Mg^{2+}]$, the low-crosslinking 'kinase-off' state of the sensor becomes dominant, and the corresponding activity of the autokinase will be dictated by the autokinase equilibrium when coupled to this 'kinase-off' state, $K_{AK}$. At low $[Mg^{2+}]$, the high-crosslinking 'kinase-on' state of the sensor will be dominant, and the corresponding activity of the autokinase will be dictated by α*$K_{AK}$. The midpoint of transition will depend on the relative magnitudes of all the parameters. However, the range of behaviors possible by this model of coupling depends heavily on the intrinsic equilibria of the sensor and autokinase themselves ($K_{Sen}$, $K_{AK}$). We observed that the two-domain model in *Figure 5B* captures much of the phenotypic behavior of the mutants shown in *Figure 2C*. However, different effects were observed for decoupling before and after the HAMP domain (*Figures 3 and 4*) indicating that it needs to be treated as a separate domain with its own equilibrium constant and independent coupling to both the sensor and catalytic domains.

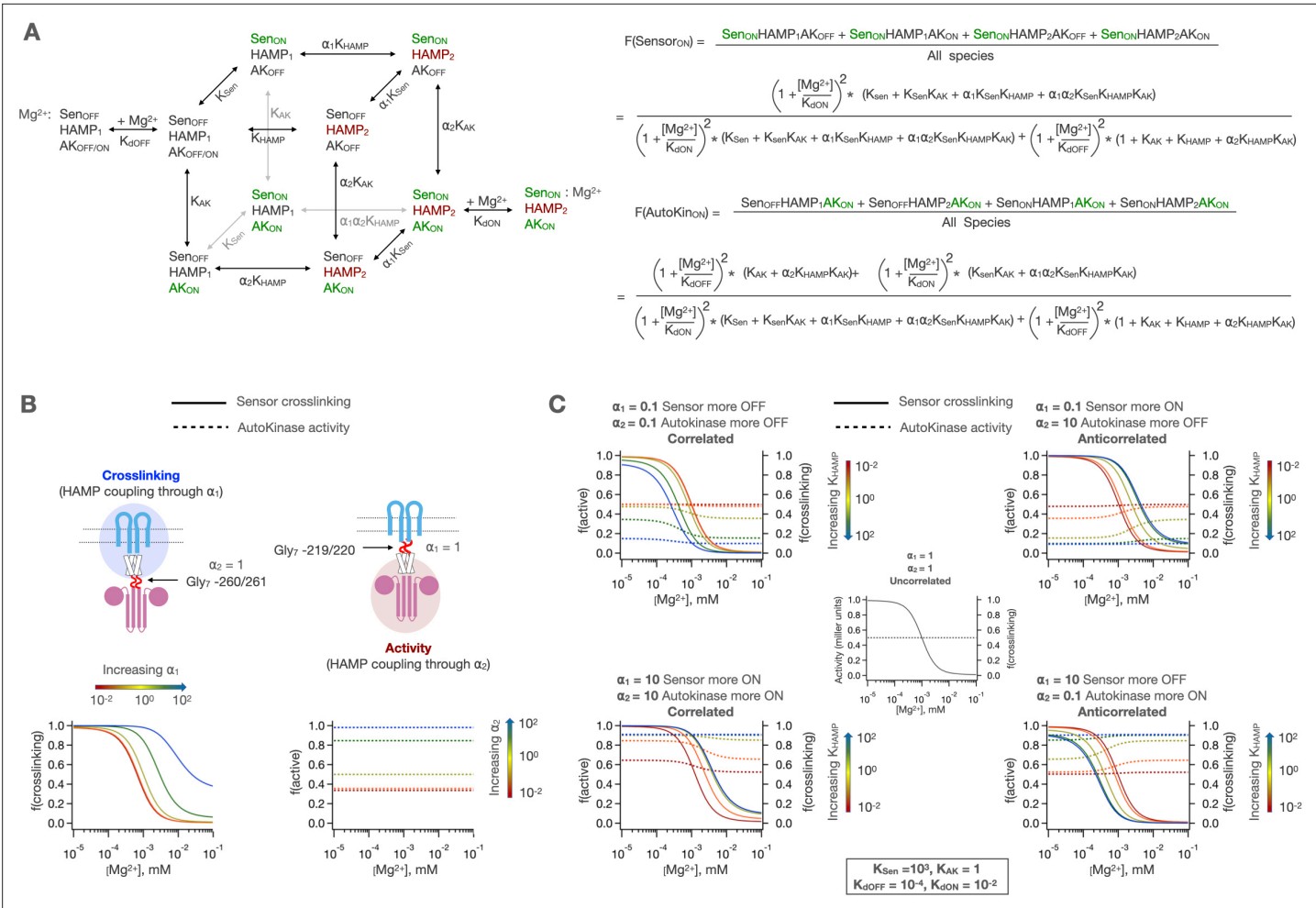

**Figure 6.** 3-domain allosteric coupling model for PhoQ signaling. (**A**) The HAMP domain is allowed to sample a two-state equilibrium between 'HAMP$_1$' and 'HAMP$_2$' states with the equilibrium constant $K_{HAMP}$. The sensor and autokinase domains of PhoQ are allowed to sample both 'kinase-on' and 'kinase-off' signaling states while coupled to 'HAMP$_1$' state in adjacent HAMP domain with equilibria $K_{Sen}$ and $K_{AK}$, respectively. When adjacent states are in 'kinase-on' or 'HAMP$_2$' states, the equilibria for transition are scaled by $\alpha_1$ (sensor-HAMP) or $\alpha_2$ (HAMP-autokinase). Predicted fraction of sensor crosslinking or autokinase activity are computed as shown to the right. Please note that Mg$^{2+}$ binding is allowed for all eight possible signaling states but are omitted except for two reference states for clarity. Similarly, three equilibria arrows and constants have been shaded grey to spatially differentiate them from nearby equilibria. (**B**) The HAMP domain allows for the independent modulation of the basal state of the sensor or autokinase. When $\alpha_2 = 1$, the HAMP domain modulates the [Mg$^{2+}$] dependent transition of the sensor, and when $\alpha_1 = 1$, the HAMP domain modulates the basal activity level of the autokinase. (**C**) The two allosteric coupling constants allow for both correlated and anticorrelated modulation of sensor and autokinase and allow for potentiation of both the 'kinase-on' and 'kinase-off' states.

## Three-domain allosteric coupling mechanism of signal transduction

Based on the results of Gly$_7$ insertion mutants and our inability to fully explain our data set with two-domain model, we developed a three-domain model, with allosteric couplings defined before and after the HAMP domain. In this model, the HAMP domain has its own intrinsic equilibrium, $K_{HAMP}$, and there are two coupling constants that describe how the sensor couples to the HAMP domain ($\alpha_1$), and how the autokinase couples to the HAMP domain ($\alpha_2$). All possible state transitions are enumerated in *Figure 6A*. This treatment allows for semi-independent modulation of the sensor and autokinase using the intrinsic equilibrium of the HAMP domain in the following ways. In the case where $\alpha_2 = 1$, the autokinase is decoupled from the sensor+ HAMP. In this scenario, the HAMP domain can modulate the [Mg$^{2+}$] dependent state transition of the sensor through coupling via $\alpha_1$ without altering the basal autokinase activity, as shown in *Figure 6B*. In the case where $\alpha_1 = 1$, the sensor is decoupled from the HAMP+ autokinase, and the HAMP domain can modulate the basal (and ligand-insensitive) activity of the autokinase through coupling via $\alpha_2$, as shown in *Figure 6B*. When the protein is fully coupled

(i.e. $\alpha_1$, $\alpha_2 \neq 1$), we can potentiate the 'kinase-on' or 'kinase-off' states of the sensor and autokinase in a manner that depends on both $K_{HAMP}$ and $\alpha_n$'s, as shown in *Figure 6C*. Of particular interest is the case where $\alpha_1$, $\alpha_2 <1$, which enables the simultaneous potentiation of the 'kinase-off' state, while maintaining the correlated sensor-autokinase behavior of PhoQ. This matches our observation that the 'kinase-off' states of both the sensor and autokinase were potentiated by the HAMP domain in our Gly$_7$ insertion experiments (*Figure 3*). Other possible behaviors with this 3-domain model include correlated sensing with 'kinase-on' potentiation, and anticorrelated signaling.

In order to fit our semi-empirical models to experimental observations, we generated a set of 35 single-point mutants and Gly$_7$ insertions and *simultaneously* determined the sensor-crosslinking and autokinase activity at five different concentrations of Mg$^{2+}$. In addition to point-mutations along the signal transduction pathway, we inserted Gly$_7$ sequences between TM and HAMP domains (Gly$_7$ 219/220), as well as between the HAMP and autokinase in two locations (Gly$_7$ 260/261, Gly$_7$ 270/271) to disrupt interdomain coupling. We chose to insert Gly$_7$ both before and after the S-Helix motif (res 261–270) since this entire region is considered a coupling motif between the HAMP and Autokinase. We also included Gly$_4$ insertion at 260/261 which suffices as an alternate insertion for decoupling HAMP from autokinase. Finally, we combined Gly$_7$ insertions with some point mutations that show behavior markedly different from WT (S217W, E232A, N255A, Y265A) to further differentiate between changes in domain two-state equilibria and changes in interdomain allosteric coupling.

Using this set of mutants, we next determined the five core allosteric parameters ($K_{Sen}$, $K_{HAMP}$, $K_{AK}$, $\alpha_1$, $\alpha_2$), and the dissociation constants for Mg$^{2+}$ to the two sensor states ($K_{dOFF}$, $K_{dON}$, *Figures 6A and 7A*). One last parameter (S) is a scaling factor that relates the mole fraction of autokinase in the 'kinase-on' state to the experimentally observed Miller units associated with the beta-galactosidase transcription, which were obtained under strictly controlled experimental conditions to assure uniformity between mutants. In all, we sought to determine eight constants for each mutant. However, given the spacing of the points in our dose-response curves, it is only possible to obtain three pieces of information, that is, the top, bottom, and midpoint of the curves. Thus, with only six pieces of information (three each from crosslinking and transcriptional activation) for each mutant, the model is under-determined for any one mutant. We avoid this problem by using global fitting. For a given mutant, only one or two (or in a single occasion, three) of the parameters are allowed to vary, with the others being fit as global parameters that are shared with other mutants. The choice of which parameters to vary is determined by the location of the perturbation on the primary sequence of PhoQ (*Figure 7—figure supplement 1*, see methods). For example, a mutation near the N-terminus of the HAMP domain would be expected to primarily alter $\alpha_1$ and $K_{HAMP}$, so these values were allowed to vary locally. Mutations near the center of the tertiary structure of the HAMP domain are allowed to vary $K_{HAMP}$ alone and so on. This results in an overall fit with 62 adjustable parameters corresponding to eight global parameters, 47 locally varied parameters, and seven parameters fixed to a value of 1 to account for Gly$_7$ insertions (*Table 1*). By comparison, there are 6 * 36 = 216 observables. Thus, in theory, the data should be more than sufficient to define the independent parameters.

This model was globally fit using our mutant dataset as explained in detail in the Materials and methods section. Briefly, we standardize the ranges of autokinase activity measurements (Miller units from beta-galactosidase assay) by the global average activity in our dataset. This normalizes the range of autokinase activity to one that is similar to crosslinking fractions (range 0–1) and gives both types of experimental measurements similar weights in our global fits. We give additional weight to data with experimental replicates (and hence greater certainty) by simply treating each replicate as an independent data set, with all the variables held constant between replicates during fit. Each parameter is allowed to sample a 10-log range of possible values, and the best fit is determined by minimizing the sum of residuals across the entire dataset. In order to avoid getting trapped in any local minima of the parameter space, we repeat the fit 125,000 times using randomly generated starting values for each parameter and determine confidence intervals for our parameters using bootstrapping to generate over 3000 synthetic dataset fits (see Materials and methods for details). Where mutations or insertions have been introduced, we allow the parameters expected to be affected by the mutation to vary locally for the corresponding data set. Moreover, six mutants can be fit with fewer local parameters than were utilized in the fit, as the values for some of these locally fit parameters remain close to the globally fit value (within 10%), as highlighted in *Table 1* (green).

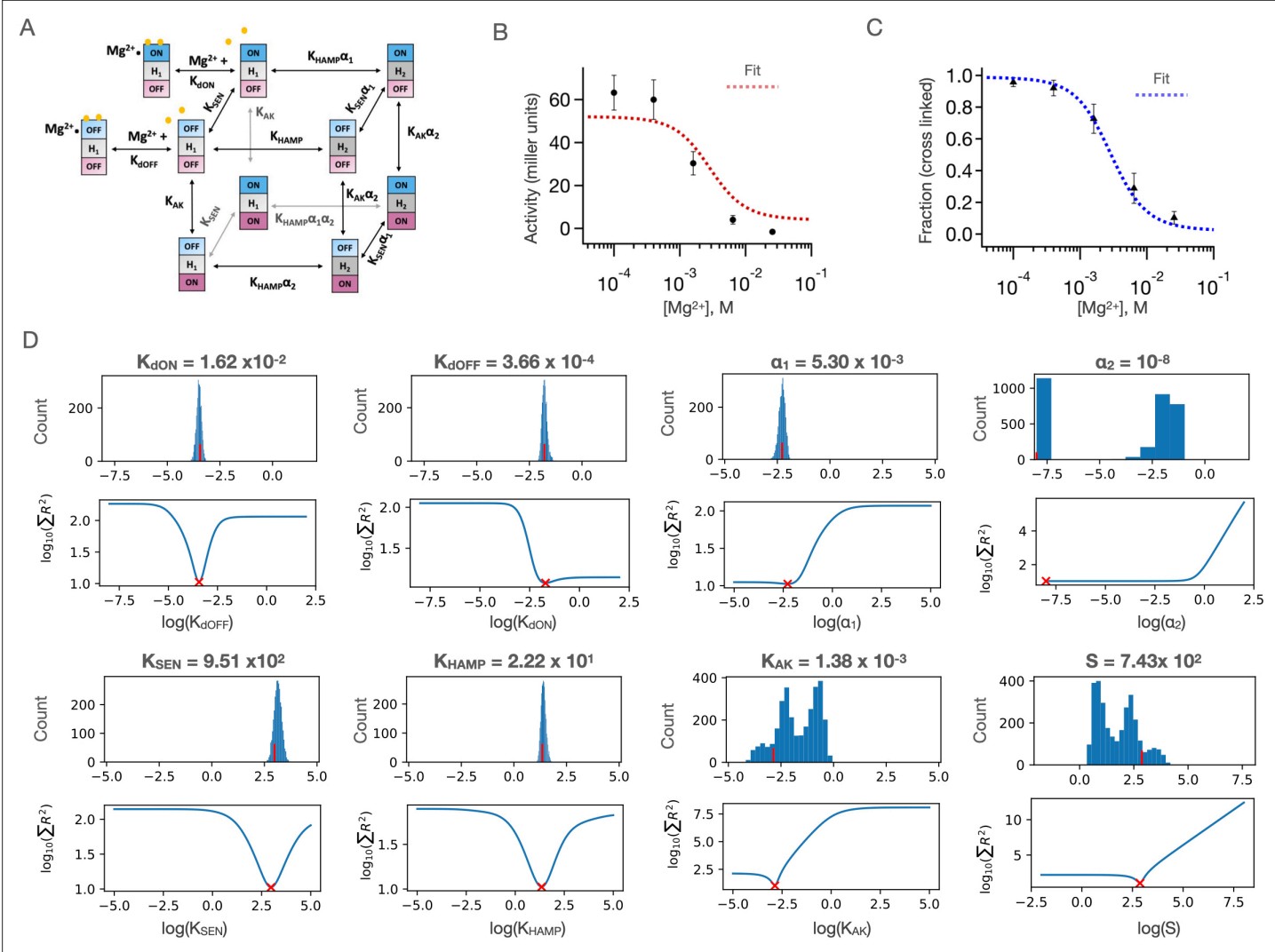

**Figure 7.** Results of three-domain two-state allosteric model fit of PhoQ activity. (**A**) Three-domain two-state allosteric model used for fitting (see also *Figure 6A*) (**B**) Fits to the [$Mg^{2+}$]-dependent kinase activity and (**C**) sensor crosslinking for 'wild type' Y60C PhoQ are shown. Error bars correspond to± SD for n = 9 biological replicates. (**D**) Bootstrapped confidence intervals (top) and residual sweep analyses (bottom) are shown for all eight global parameters. The value of the fit is indicated with red (**x**) and (|) marks. The confidence intervals of parameters S, $K_{AK}$ and $\alpha_2$ are further parsed in *Figure 7—figure supplement 2*, and the confidence intervals for $\alpha_2$ are further parsed in *Figure 7—figure supplement 3*.

The online version of this article includes the following source data and figure supplement(s) for figure 7:

**Figure supplement 1.** Point mutations and Gly7 insertions in PhoQ.

**Figure supplement 2.** Effect of constraining S and $K_{AK}$.

**Figure supplement 3.** Effect of constraining $\alpha_2$.

**Figure supplement 4.** A two-domain two-state allosteric coupling model does not fit set of PhoQ mutants.

**Figure supplement 4—source data 1.** Two-domain two-state model fitting parameter values for PhoQ mutant dataset.

We are able to obtain a remarkably good fit for our entire dataset with the aforementioned considerations. *Figure 7* shows the results of the best obtained fit for 'wild type' PhoQ (Y60C) sensor-crosslinking (*Figure 7B*) and autokinase activity (*Figure 7C*). Since the WT data were fit entirely globally, they represented the most stringent test for the performance of our model overall, and qualitatively showed good agreement between model fit and experimental data. The values of the eight global parameters corresponding to this wild type fit are shown in *Figure 7D*, alongside two metrics of fit quality. The first metric is a bootstrapped confidence interval, with the frequency histogram of resulting fit values shown in the top panels. The second metric is a parameter sweep analysis in which

**Table 1.** List of mutant parameter fits.

Local parameters whose values remained within 10% of the global fit value are highlighted in bold font and green background. Parameters whose value drifted to one end of the explored parameter range are highlighted in italicized font and orange background. Key: 'TM7' → $Gly_7$ insertion at 219/220; 'HAMP 4' → $Gly_4$ insertion at 260/261; 'HAMP 7' / 'H7' → $Gly_7$ insertion at 260/261; 'SH7' → $Gly_7$ insertion at 270/271.

| Mutation | $K_{Sen}$ | $K_{HAMP}$ | $K_{AK}$ | $\alpha_1$ | $\alpha_2$ | S | $K_{dOFF}$ | $K_{dON}$ |
|---|---|---|---|---|---|---|---|---|
| Y60C | 9.5 E + 02 | 2.2 E + 01 | 1.4 E –03 | 5.3 E –03 | *1.0 E –08* | 7.4 E + 02 | 3.7 E –04 | 1.6 E –02 |
| Y60C HAMP 4 | | 4.5 E + 01 | | | 1.0 E + 00 | | | |
| Y60C HAMP 7 | | | | | 1.0 E + 00 | | | |
| Y60C SH7 | | **2.1 E + 01** | 7.7 E –04 | | 1.7 E + 00 | | | |
| Y60C TM7 | | | | 1.0 E + 00 | | | | |
| Y40W | 1.4 E + 03 | | | | | | | |
| S43W | 3.8 E + 02 | | | | | | | |
| E55A | 4.1 E + 02 | | | | | | | |
| E55S | 1.5 E + 03 | | | | | | | |
| V191W | 1.2 E + 03 | | | | | | | |
| I207A | 6.9 E + 02 | | | 1.1 E –01 | | | | |
| L210A | 3.1 E –03 | | | 9.1 E + 04 | | | | |
| A213W | 4.0 E + 04 | | | *1.0 E –05* | | | | |
| S217W | | 7.1 E –01 | | 7.6 E –01 | | | | |
| S217W + H7 | | | | | 1.0 E + 00 | | | |
| S217W + TM7 | | | | 1.0 E + 00 | | | | |
| I221F | | **2.0 E + 01** | | 1.5 E –01 | | | | |
| L224A | | 1.6 E + 01 | | 1.3 E –01 | | | | |
| L224F | | 6.8 E + 01 | | 1.2 E –02 | | | | |
| A225F | **1.0 E + 03** | **2.4 E + 01** | | 2.3 E –01 | | | | |
| E232A | | 1.1 E + 02 | | | 1.4 E + 00 | | | |
| E232A + H7 | | | | | 1.0 E + 00 | | | |
| E233A | | **2.3 E + 01** | | | *1.0 E –08* | | | |
| R236A | | 8.6 E + 00 | | | | | | |
| N240A | | 1.7 E + 01 | | | | | | |
| R245F | | **2.2 E + 01** | | | | | | |
| L254A | | | *1.0 E –05* | | *1.0 E –08* | | | |
| N255A | | | **1.3 E –03** | | 4.9 E –01 | | | |
| N255A + H7 | | | | | 1.0 E + 00 | | | |
| R256A | | 3.6 E + 01 | | | *1.0 E –08* | | | |
| L258A | | | *1.0 E –05* | | *1.0 E –08* | | | |
| E261F | | 3.9 E + 01 | | | 9.9 E –01 | | | |
| Y265A | | | 4.1 E –04 | | 3.2 E + 00 | | | |
| Y265A + TM7 | | | | 1.0 E + 00 | | | | |
| Y265A + SH7 | | | | | 3.8 E + 00 | | | |
| R269L | | | 3.3 E –04 | | *1.0 E –08* | | | |

the global sum of residuals is evaluated as the value of the indicated parameter is allowed to vary while all other parameters are held fixed. Five of our global parameters, $K_{dOFF}$, $K_{dON}$, $K_{Sen}$, $K_{HAMP}$ and $\alpha_1$ show excellent convergence to the 'best fit' value, with well-defined minima in the sum of residuals as we explore parameter value. Three parameters, $K_{AK}$, S and $\alpha_2$ show strong signs of covariability, and wider confidence intervals. In the fully activated state, the observed signal is defined by the product of the scaling factor, S, and the fraction of the protein in the 'kinase-on' state (approximately $S*K_{AK}$). This product is well-defined and converges to a value of $\approx$ 1.02. However, as $K_{AK}$ is lowered below this value, S increases in parallel to maintain a constant value for the product of $S*K_{AK}$. In *Figure 7—figure supplement 2*, we show that when the values of S are restrained, the values of $K_{AK}$ are also restrained, and vice versa. Nevertheless, we can place a functionally meaningful upper limit on $K_{AK}$, of approximately 0.1. Similarly, we can place an upper limit on the value of 0.1 for $\alpha_2$, which represents the negatively cooperative coupling of $K_{AK}$ to the parameters defining the other domains. $\alpha_2$ describes the difficulty of the autokinase to transition into the 'on-state' when coupled to the $HAMP_2$ state vs the $HAMP_1$ signaling state. Our data show that this transition is disfavored. *Figure 7—figure supplement 3* shows that for both peaks of $\alpha_2$ values centered around $10^{-2}$ and $10^{-7}$, we converge to similar parameter fits for the other global parameters since both values of $\alpha_2$ establish a tightly coupled 'kinase-off' state within the sensitivity of our experiments. Therefore, these uncertainties do not affect any of our conclusions below, which depend on presence of strong versus weak and negative versus positive coupling. We also examined the ability to fit a simpler two-domain allosteric coupling model (*Figure 5B*, see Materials and methods *Equation 2*) to our data. This model failed to globally fit the set of sensor crosslinking and kinase activities of WT PhoQ, point mutants and $Gly_7$ insertions (*Figure 7—figure supplement 4*).

One feature that was somewhat surprising was that $K_{AK}$ was unfavorable towards forming the 'kinase-on' versus 'kinase-off' states ($K_{AK}$ <1), even at limiting low concentrations of $Mg^{2+}$. This indicates that the observed activity for the WT protein is less than what is observed for some of the mutants, and what might be observed in a hypothetical state in which the autokinase is unfettered by connections to HAMP and the membrane. Although unexpected, this finding is consistent with a large body of data (*Fernández et al., 2019*; *Wang et al., 2015*; *Koh et al., 2021*), and has been observed in PhoQ with antimicrobial peptide stimulation (*Matamouros et al., 2015*). Thus, in ligand-responsive HKs, evolution does not drive toward maximal activity which might lead to wasteful and toxic transcription, but instead a finely tuned value that is titrated to the degree of transcription required for function.

## Application of the three-domain model to a set of mutants illustrates how substitutions distant from active sites modulate signal strength and ligand sensitivity

The values of the parameters provide a detailed view of the energy landscape of PhoQ, in the 'kinase-on' and 'kinase-off' state – and how it is modulated by binding to $Mg^{2+}$ and mutations. The parameters are consistent with our observations that the HAMP domain is a significant modulator of the intrinsic equilibria of the sensor and autokinase domains. At high $Mg^{2+}$ concentrations, PhoQ is in a 'sensor-off' and 'autokinase-off' state. With respect to this reference 'kinase-off' state, the HAMP domain has a thermodynamically favored signaling state, '$HAMP_2$', with a fit equilibrium value of $K_{HAMP}$ = 22. This favored state of the HAMP domain is more strongly coupled to these 'kinase-off' states and serves to dampen the otherwise favorable transitions of both the sensor and autokinase domains to the 'kinase-on' conformation. The sensor's propensity to switch to a 'sensor-on' state is reduced from a highly preferred equilibrium $K_{Sen}$ = 9.5 x $10^2$, to a modest downhill equilibrium of $\alpha_1*K_{Sen}$ = 5.0 when the HAMP domain is in this $HAMP_2$ state. This latter equilibrium is weak enough to be overcome by $Mg^{2+}$ binding, and the 'sensor-off' state is further stabilized with more ligand binding. The '$HAMP_2$' state that is preferred in this 'sensor-off' state is also strongly coupled to the 'kinase-off' state of the autokinase, reducing the propensity of the autokinase to switch to the 'kinase-on' state from $S*K_{AK}$ = 1.0 to $\alpha_2*S*K_{AK} \leq 10^{-3}$. Thus, the $HAMP_2$ state behaves as a negative modulator of the intrinsic propensities of the sensor and autokinase. At high enough $[Mg^{2+}]$, the entire population ensemble is predominantly in the $sensor_{OFF}$-$HAMP_2$-$Autokinase_{OFF}$ state.

In the absence of ligand, the sensor's modest downhill equilibrium to the 'kinase-on' state is strongly tied to a switch of the HAMP domain from '$HAMP_2$' to '$HAMP_1$', with an equilibrium = 1/

($\alpha_1 K_{HAMP}$) = 25. The HAMP$_1$ state is weakly coupled to the autokinase, which allows the autokinase to sample both kinase-off and kinase-on state, with an effective equilibrium of S.$K_{AK}$ = 1.02. This allows for the partial decoupling of the sensor and the autokinase at low Mg$^{2+}$ concentrations observed in wild type PhoQ (*Figure 2C*) with the population ensemble composed of both sensor$_{ON}$-HAMP$_1$-Autokinase$_{ON}$ and sensor$_{ON}$-HAMP$_1$-Autokianse$_{OFF}$ states. Because the HAMP$_1$ state is weakly coupled to both adjacent domains, the equilibria constants $K_{Sen}$ and $K_{AK}$ are close to the intrinsic equilibria of these domains when uncoupled from the HAMP domain altogether. In other words, $K_{Sen}$ = 9.5 × 10$^2$ reflects the high propensity of the sensor to switch to the 'sensor-on' state when uncoupled from the HAMP domain, and S.$K_{AK}$ = 1.0 reflects the propensity of the autokinase to have as high an activity as WT PhoQ at low [Mg$^{2+}$], as shown earlier with Gly$_7$ disconnections in *Figure 3*.

The parameters for individual mutations show how amino acid substitutions alter the energy landscape and how these changes in turn alter the phenotype. Before discussing the effects of substitutions, however, it is important to address the overall quality of the fit over the full ensemble of mutants. *Figure 8* shows the results of fits for our mutations and Gly$_7$ insertions with mutations color-labeled according to the locally varied parameters as in *Figure 7—figure supplement 1*. The corresponding fit values are listed in *Table 1*, and confidence intervals and parameter sensitivity analyses are shown in *Figure 8—figure supplement 1*. While it is possible to construct models with even more states, for example, by treating the TM domain separately rather than as an extension of the sensor domain, the number of parameters – and their uncertainty- rapidly increases. We therefore chose the simplest model required to describe the entire set of data.

We obtained fits within experimental error for the [Mg$^{2+}$]-dependent transcriptional activity of our entire mutant data set. Thus, the model worked well for all ligand-sensitive mutants. The only deviations lay in the crosslinking data for non-functional mutants that were decoupled in the transcriptional output (*Figure 8—figure supplement 2*). One such set of mutants (I221F, L224A, and A225F) had substitutions at the C-terminal end of the second TM helix. While the midpoint and lower limit were well described by the model, the experimentally observed extent of crosslinking reached an upper limit of 65–80% crosslinking at low Mg$^{2+}$, less than the predicted value near 100%. Given the location of the substitutions near the membrane interface, it is possible that a portion of the protein is not fully inserted and hence the samples used for western analysis might have been contaminated by cytoplasmically localized, and not yet membrane-inserted protein, which would be expected to remain not cross-linked. There are also two mutations localized near the interface between the HAMP and the autokinase domains where the mid-point is poorly fit (L254A, L258A), potentially owing to our choices of parameters to locally float for these mutants (*Figure 8—figure supplement 2D*-**E**) as discussed in Materials and methods. Significantly better fits were obtained by altering the parameters varied for these mutants from $K_{HAMP}$ and $\alpha_2$ to $K_{AK}$ and $\alpha_2$. Thus, these residues may be involved in the underlying equilibrium of the autokinase domain itself due to their proximity to the 'S-helix' that connects the HAMP domain to the autokinase. Finally, double mutants are not fit well, especially when the two sites of mutation are in close proximity (S217W + HAMP Gly$_7$, N255A + HAMP Gly$_7$, Y265A + Sensor Gly$_7$, *Figure 8—figure supplement 2F*-**H**). This is likely because the thermodynamic effects of double mutations are often non-additive in structurally and sequentially proximal positions that interact directly. Additionally, while we observe relatively invariant expression of almost all variants (as seen in the western analysis used to quantify crosslinking), some variants, particularly double mutants required slight induction of expression with 10 µM IPTG for observable levels of membrane-inserted PhoQ by western-blotting (see Materials and methods). In summary, the crosslinking and transcriptional activity data are very well fit for the entire set of mutants, except for a fraction of the nonfunctional mutants in which Mg$^{2+}$ binding and transcription were significantly decoupled. Even for these mutants, however, there is a qualitative fit to the data, and possible reasons for the deviation.

Our results illustrate how allosteric coupling of domain equilibria changes in response to single-site substitutions. Although we chose a collection of mutations that were not involved in Mg$^{2+}$ binding and catalysis, we observed a large range of effects on the transcriptional response of the mutants, including an inverse response in E232A and Gly$_7$ 270/271 insertion. The advantage of the current analysis is describing how these mutations alter the energetics of individual domains, and their coupling to adjacent domains. As an organism evolves to match its environment, its sensory systems need to adjust to the ligand-sensitivity (midpoint of the dose-response curve), the magnitude of the increase

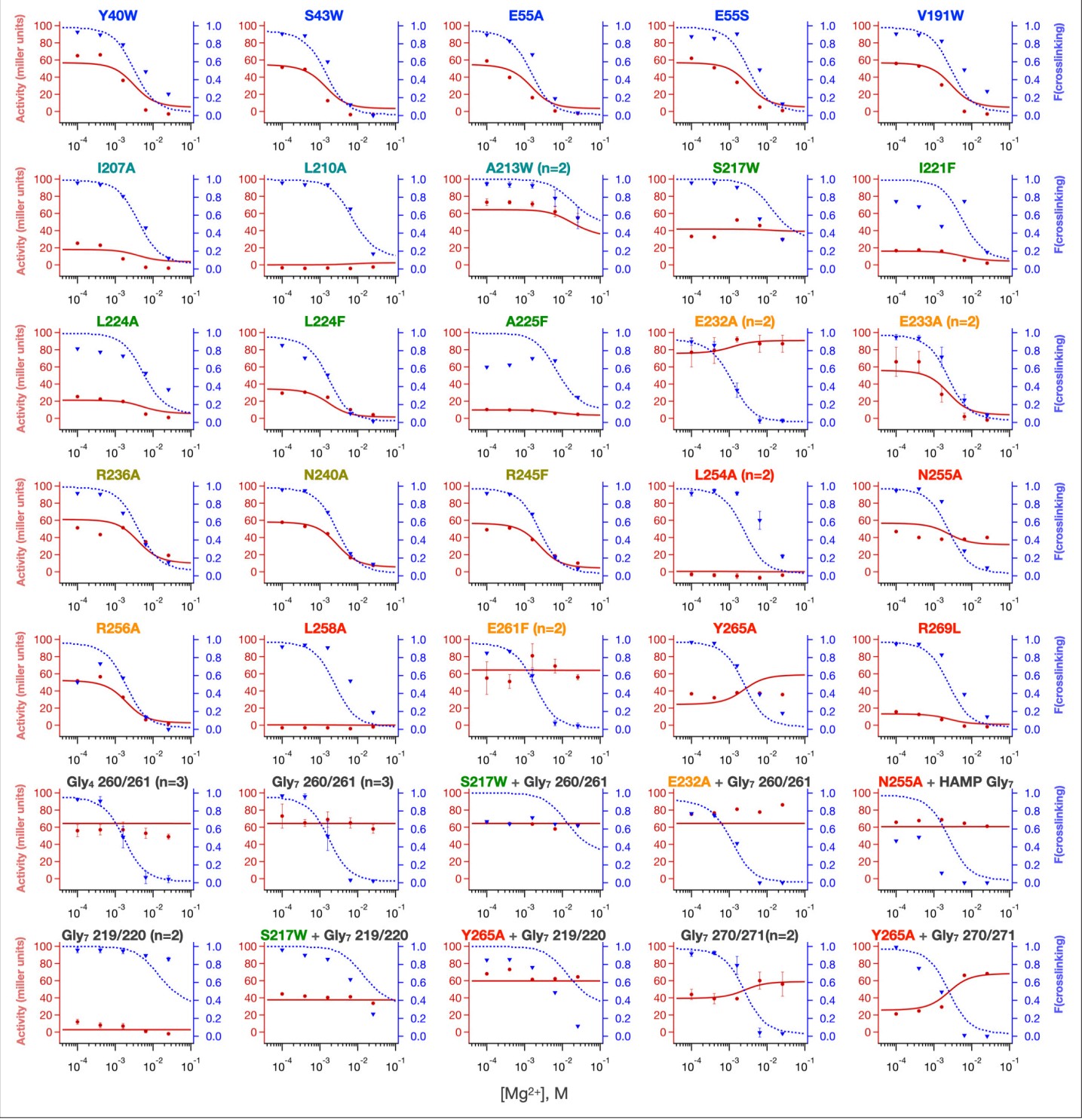

**Figure 8.** Local fits of sensor crosslinking and kinase activity for 35 PhoQ mutations. Fits to activity (red line, closed circles) and sensor crosslinking (blue dashed line, triangles) are shown for the entire PhoQ dataset. The color of mutations matches the color scheme in *Figure 7—figure supplement 1* to indicate locally varied parameters, and these parameters are listed in Table 1 and Table 2. Confidence intervals and residual sweep analyses are presented in *Figure 8—figure supplement 1*. Poor fits are highlighted in *Figure 8—figure supplement 2*.

The online version of this article includes the following source data and figure supplement(s) for figure 8:

**Source data 1.** [Mg2+]-dependent activity and sensor crosslinking of PhoQ mutants.

**Figure supplement 1.** Bootstrapped confidence intervals and residual sweep analyses for PhoQ mutant fits.

**Figure supplement 2.** Poor fits were obtained for crosslinking at low [Mg$^{2+}$] for (**A**) I221F, (**B**) L224A and (**C**) A225F.

in the response (in this case, the activity in the absence of $Mg^{2+}$/activity in presence of $Mg^{2+}$) and the basal activity (in the presence of saturating $Mg^{2+}$). We consider these features separately.

In a well-coupled system such as WT PhoQ, the midpoint can be modulated by point mutations anywhere along the signal transduction pathway between the sensor and the autokinase. The only requirement is for the substitution to have an effect on the internal equilibrium constant for the kinase-promoting versus the phosphatase-promoting conformations of the domain that houses the mutation. So long as the domains are tightly coupled, then an n-fold change in the internal equilibrium will translate to an n-fold change in the midpoint of the overall dose-response curve. Moreover, as the couplings $\alpha_1$ and $\alpha_2$ become less strong, the magnitude of the shift in the dose response curve is decreased. Thus, it is not necessary to change the binding interactions with the metal ions to affect changes in the ligand sensitivity of the system which provides the system a wealth of opportunities to tune sensitivity.

The fractional change in the kinase activity that can be achieved upon saturation of the ligand-binding sites is a second factor, which ranges with the requirements of a system. For example, the change in transcriptional response in PhoQ is modest, reaching about a factor of 5–20-fold change, while other two-component systems such as VirA have a dynamic range as large as $10^5$ (*Wang et al., 2015*; *Baruah et al., 2004*; *Eguchi and Utsumi, 2014*; *Gao and Lynn, 2005*). It is however possible that combinatorial signals might further increase the overall kinase activity of PhoQ from that observed at low $Mg^{2+}$, owing to additive modulation of the sensor domain by other stimuli $H^+$, antimicrobial peptides, MgrB, SafA, UgtL *Yoshitani et al., 2019*, as some independence between signal inputs has been demonstrated for *S. Typhimurium* PhoQ (*Hicks et al., 2015*). For simple systems that respond to a single ligand, the maximal response is defined by the ratio of the intrinsic affinities of the ligand for the 'kinase-on' versus 'kinase-off' conformations ($K_{dON}/K_{dOFF}$). Mutations that decrease the coupling attenuate the maximal response, and the system becomes decoupled when $\alpha_1$ or $\alpha_2$ reaches 1. The maximal response in absolute terms is another factor, which depends on the kinetic efficiency of the underlying autokinase domains. When untethered from the remainder of the protein, the autokinase domain shows large increases activity (*Figure 3*), so the role of the remainder of the protein can be seen as a negative regulation. Indeed, we find that $K_{AK}$ is significantly less than 1, and this value can be positively modulated by some mutations that reach transcriptional levels somewhat greater than WT for PhoQ. In summary, there is a diversity of mechanisms that nature can call upon to alter the activity of HKs, as illustrated in a relatively small sampling of the 35 mutants studied here.

## Discussion

It has been appreciated for several decades that the effector domains of multi-domain signaling proteins can produce responses that are either potentiated or diminished relative to the change in state of sensory domains that drive these responses. S J Edelstein and J P Changeux in seminal

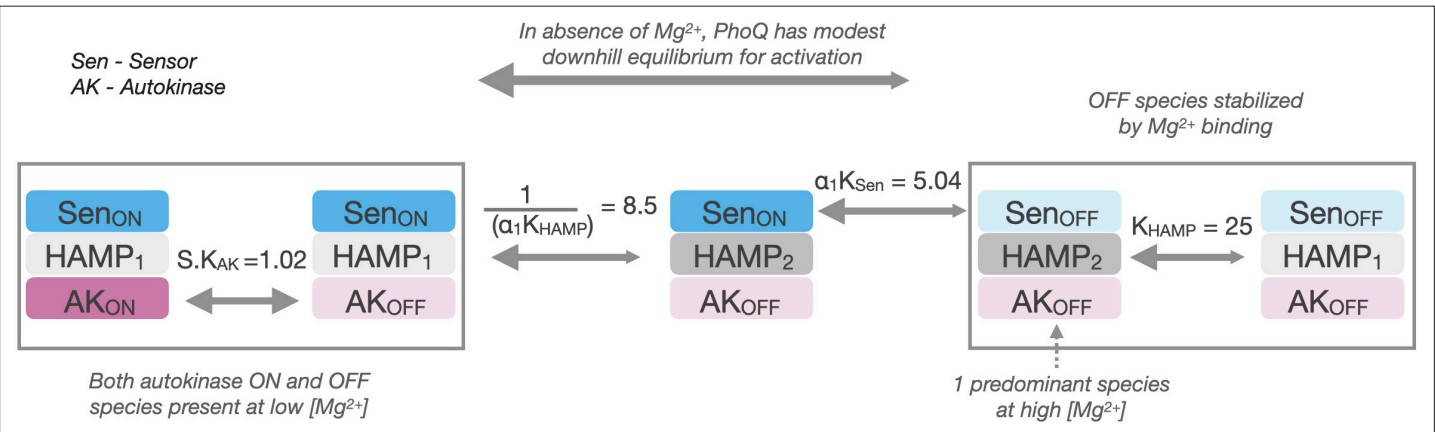

**Figure 9.** Allosteric pathway for PhoQ activation. In the absence of $Mg^{2+}$, PhoQ has a moderate downhill equilibrium to a mixture of active states. $Mg^{2+}$ binding is sufficient for overpowering this equilibrium and stabilizing the 'kinase-off' state, resulting in a predominantly Sensor-off/HAMP2/Autokinase-off population.

work coined the terms 'hyper-responsive' and 'hypo-responsive' to define this uncoupled behavior between domains (*Edelstein and Changeux, 1996*), which has since been examined in other classes of multi-domain signaling proteins such as GPCRs (*Smith et al., 2018*; *Changeux and Edelstein, 2005*; *De Lean et al., 1980*). In this work, we examine the coupling behavior between the sensor and effector domains of transmembrane bacterial sensor histidine kinases and possible roles of modularly inserted signal transduction domains in optimizing this coupling behavior using a model gram negative HK, PhoQ. We find that the intervening HAMP signal transduction domain is necessary to assemble an overall bistable histidine kinase from $Mg^{2+}$-sensor and autokinase-effector domains that are too biased to one signaling state ('kinase on' state). This is accomplished by strongly coupling the thermodynamically preferred state of the HAMP domain to the disfavored 'kinase off' signaling states of sensor and autokinase, opposing these otherwise strong equilibria such that the overall assembly is bistable and significantly modulated by ligand binding (*Figure 9*). Thus, the HAMP domain does more than transmit the response; it instead serves to tune the ligand-sensitivity amplitude (i.e. minimum and maximum signal and midpoint of transition) of the response.

Evolutionarily, the insertion of signal transduction domains in HKs could allow for the facile modulation of the intrinsic equilibria of sensor and effector domains and their coupling behavior, which may be more difficult to alter through the direct mutation of these domains themselves. The sequence and subsequent structures of sensors and autokinase domains are subject to many evolutionary constraints, be it the specificity and affinity for ligands in sensor domains, the specificity for membrane homodimerization of HKs (*Ashenberg et al., 2011*), or the cognate specificity for response regulator (*Casino et al., 2009*; *Podgornaia et al., 2013*; *Skerker et al., 2008*; *Buschiazzo and Trajtenberg, 2019*; *Ohta and Newton, 2003*) and the ability to inhabit and switch between the various conformations required for a full catalytic cycle in the autokinase domain (*Jacob-Dubuisson et al., 2018*). Furthermore, most two-component systems feature multiple accessory protein components involved in sensing, feedback regulation and cross-talk with other signaling systems, which add evolutionary constraints to these domains (*Salazar and Laub, 2015*). In the closely related class of chemotaxis proteins, the analogous transmembrane protein is also subject to extensive covalent modifications that modulate activity. When all these evolutionary activity and specificity considerations are met, the resulting domain may not be ideally bi-stable in isolation. Indeed, in PhoQ, we find that both sensor and autokinase highly prefer the 'kinase-on' state, and therefore cannot be allosterically connected to make an overall bistable protein capable of being converted to the 'kinase-off' state by $Mg^{2+}$ binding. The presence of one or more signal transduction domains allows for two advantageous considerations for producing and fine-tuning overall HK bistability; the thermodynamic stability of the signal transduction domain can be used to preferentially stabilize or destabilize a given signaling state of sensors or autokinases indirectly through allosteric coupling, and the strength and even direction of coupling can be easily modulated through mutations at the domain junction, rather than mutations that may alter the core functions of the sensor/autokinase themselves. In the case of PhoQ, the HAMP is an allosteric repressor of autokinase activity; as such, mutations that destabilize the thermodynamically preferred HAMP state ($HAMP_2$) or reduce its coupling to the autokinase would allosterically increase kinase activity and vice versa.

The latter phenomenon is especially potent in the context of alpha-helical coiled-coil connections between domains of HKs, in which a drastic change in coupling or thermodynamic stability can be caused by minor sequence insertions, deletions and alterations due to the highly regular and cooperative nature of coiled-coil stabilizing interfaces. We have shown that the insertion of a stretch of glycine residues is sufficient to almost completely uncouple domains. On the other extreme, a well folded coiled coil junction can create strong allosteric coupling due to the cooperative folding and stability of such a motif. A range of stabilities can be achieved by various means, including the insertion or deletion of one or more residues to disrupt the canonical heptad pattern of hydrophobic residues of the dimeric core of the protein, as is often observed in the conserved S-helix motif, which connects HAMP to autokinase domains in HKs (*Anantharaman et al., 2006*). *Schmidt et al., 2017* showed that crystal structures of cytoplasmic domains in different conformations accommodate the structural deviations of these S-Helix sequence insertions by delocalizing the strain over different lengths of the proximal alpha-helical core. These different 'accommodation lengths' could be analogous to the different strengths of allosteric coupling depending on the signaling states of the adjacent domains in our equilibrium signaling model. We also find conservation of glycine motifs and helix-disrupting proline

residues in the juxta-membrane regions of chemotaxis proteins and HKs, respectively (*Fernández et al., 2019*; *Motz and Jung, 2018*; *Akkaladevi et al., 2018*), which hint at the significant modulation of allosteric coupling strength by the alteration of helical and coiled-coil geometries. In some systems, domains are even segregated to entirely different proteins, in which case the strength of the protein-protein interaction between components can be altered to vary allosteric coupling. These are all evolutionarily accessible solutions to fine-tune the function of a histidine kinase.

Finally, this evolutionary argument may also explain the lack of a parsimonious structural mechanism for signal transduction, even in HKs with a specific domain architecture. Although this problem is exacerbated by the dearth of multi-domain structures of HKs in various signaling conformation, several signaling hypotheses have been put forward regarding the structural mechanism for signal transduction in HKs, particularly in HAMP domains. These include the gear-box mechanism (AF1503, Aer2 multi-HAMP)(*Inouye, 2006*), piston mechanism (Tar) (*Chervitz and Falke, 1996*; *Falke and Erbse, 2009*), scissoring mechanism (Tar, BT4663, PhoQ) (*Molnar et al., 2014*; *Milburn et al., 1991*; *Lowe et al., 2012*), orthogonal displacement mechanism (HAMP tandems, Tar) (*Swain and Falke, 2007*; *Airola et al., 2010*; *Swain et al., 2009*) and the dynamic HAMP mechanism (Adenylate cyclase HAMP) (*Ferris et al., 2011*; *Parkinson, 2010*; *Stewart, 2014*). A recently elucidated set of structures of the sensor, TM and signal transduction domains of NarQ remains the only representative of a multidomain transmembrane structure of an HK containing a signal transduction domain, and again shows a rigid-body bending transition of the HAMP domain about the conserved N-terminal proline between apo- and holo-states of the sensor (*Gushchin et al., 2017*).

It may be that signal transduction mechanisms in HKs are as varied as their modular architecture, and many structural transitions could account for the underlying function in signaling, which is the allosteric modulation of multi-state equilibria of adjacent domains in response to structural transitions caused by a sensory event. Indeed, the only requirement for signal transduction is a series of domains with two states that either favor or disfavor the kinase state, and a means to transmit the information between the states. Helical connections between domains provide efficient coupling, but the conformational changes within the domain need not be obligatorily the same for different domains. Additionally coupling can involve tertiary contacts, which can be used in conjunction with or instead of helical connections (*Mechaly et al., 2014*; *Albanesi et al., 2009*; *Jacob-Dubuisson et al., 2018*; *Bhate et al., 2015*; *Sukomon et al., 2017*). Interestingly, the observation that PhoQ has a weakly HAMP-coupled 'kinase-on' state and a strongly HAMP-coupled 'kinase-off' state has been posited before, albeit in the context of a hypothesized tertiary contact between the membrane-distal portion of HAMP helix-1 and a loop in the autokinase domain (*Matamouros et al., 2015*). The idea that autokinase domains intrinsically have high-kinase activity and are subsequently inhibited by strong coupling to up-stream domains and the further stabilization of these inhibitory conformations by ligand-binding warrants examination as a generalizable signaling mechanism for histidine kinases.

## Materials and methods
### Materials
BW25113 and HK knockout strains were obtained from the Keio collection. TIM206 (*E. coli* Δ*phoQ*, p*mgrB*::LacZ) was obtained from Tim Mayashiro (Goulian lab). pTrc99a (GenBank # M22744) was obtained from commercial sources. pSEVA311 (GenBank# JX560331) was developed by the de Lorenzo lab and was a gift from the European Standard Vector Architecture consortium. Brilacidin was a gift from Polymedix Inc N-ethylmaleimide (NEM) was purchased from Sigma. Tris-Acetate gels (Thermofisher Scientific) and Anti-PentaHis antibody (Thermofisher Scientific) were used for western blotting.

### Methods
#### Cloning
PhoQ mutants were cloned into the pTrc99a plasmid multiple cloning site by restriction cloning. Point mutations were made by quick-change mutagenesis and confirmed by sanger sequencing. Hybrid HK-gene reporter plasmids were built in pTrc99a plasmid by introducing a c-terminally 6 x His-tagged HK construct into the IPTG inducible MCS, and the mCherry reporter sequence downstream by Gibson cloning (*Gibson et al., 2009*). Sequences of reporters are available in supplementary methods. Gly$_7$

disconnections and point mutations were introduced by a blunt-end ligation strategy and confirmed by Sanger sequencing.

## Growth of PhoQ constructs

For each biological replicate, an isolated colony of TIM206 (genotype: ΔphoQ, pmgrB::LacZ) containing various pTrc99a-phoQ constructs was grown overnight at 37 °C in MOPS minimal media + 50 µg/mL AMP and 1 mM MgSO$_4$. These overnight cultures were then diluted 50 x into 1 mL MOPS media + 50 µg/mL AMP and 1 mM MgSO$_4$ and grown at 37 °C for 2 hr. These cultures were further diluted 500 X into 30 mL MOPS minimal media + 50 µg/mL AMP containing 0.1, 0.4, 1.6, 6.4, and 25.6 mM MgSO$_4$, and grown for at least 5 hr such that the density of the culture reaches log-phase (OD$_{600}$ = 0.2–0.8). A total of 500 µL of culture is removed for evaluating beta galactosidase activity, while the remaining culture is used for western analysis. Two constructs (A225F, Y265A Gly$_7$ 260/261) showed no detectable PhoQ in membrane preparations and required induction with 10 µM IPTG during growth for observable levels of membrane-inserted PhoQ by western blot.

## Beta galactosidase activity

A total of 500 µL of PhoQ culture was combined with 500 µL of 1 x Z-buffer +40 mM beta-mercaptoethanol, 25 µL of 0.1% SDS in water, and 50 µL of chloroform in a glass culture tube and vortexed for complete lysis. The lysate was then prewarmed to 37 °C in an air incubator before addition of ONPG substrate. 0.25 mL of prewarmed 4 mg/mL ONPG in 1 x Z-buffer+ bMe was added to the lysate to initiate hydrolysis, which was then quenched with the addition of 500 µL of 1 M Na$_2$CO$_3$ after variable incubation periods. The quenched hydrolysis was then centrifuged to remove any cell debris, and absorbance at 420 nm and 550 nm was measured in triplicate using a Biotek synergy2 plate-reader with pathlength correction. Miller units were calculated as follows:

Miller units = $1000*(OD_{420} – 1.75*OD_{550})/(OD_{600}*$dilution factor*incubation time in min).

## Membrane fraction isolation

Thirty mL of PhoQ culture was centrifuged at 4350xg at 4 °C for 20 min to collect a cell-pellet. This cell pellet was immediately frozen in liquid nitrogen and stored at –80 °C until analysis. Frozen pellets were thawed, suspended and incubated on ice with 500 µg/mL N-Ethylmaleimide (NEM) and 1 mg/mL lysozyme in 50 mM TRIS buffer, pH 8, for 1 hr. Cells were then lysed by 30 seconds of tip sonication (Fisher Scientific Sonic Dismembrator Model 500, 10% Amplitude, 1 s pulse on, 1 s pulse off). Lysed cells were then centrifuged at 16,000xg for 10 min to remove cell debris. Membrane was isolated from the supernatant by further centrifugation at 90,000xg for 10 min. Membrane pellets were then resuspended in 1 X lithium dodecyl sulfate (LDS, Invitrogen) loading buffer containing 8 M urea and 500 mM NEM, boiled at 95 °C for 10 min and analyzed by western blot.

## Monomer and dimer quantification by western blot

LDS solubilized membrane prep samples were separated on 7% TRIS-SDS gels by electrophoresis at 200 V for 70 min, and then transferred onto nitrocellulose membranes by dry transfer (iBlot2). Membranes were then blocked using 1% BSA in TBS-t buffer (20 mM Tris, 2.5 mM EDTA, 150 mM NaCl, 0.1% Tween-20), probed using an anti-pentaHis HRP antibody, and visualized using luminescent ECL substrate on a BioRad imager. Bands corresponding to PhoQ monomer and dimer were quantified using Image-J software to yield a crosslinking efficiency between 0 and 1. A representative quantification of crosslinking is presented in *Figure 2—figure supplement 1*.

## Measuring activity of CpxA, BaeS

HK constructs were cloned into the MCS of pTrc99a plasmid, and the associated fluorescent reporter gene was cloned downstream. For the CpxA reporter plasmid, the response regulator CpxR, was also cloned into the MCS and transformed into AFS51 strain (ΔcpxAΔpta::Kan pcpxP::GFP) by heat shock transformation. For BaeS, the response regulator BaeR, was cloned into an additional plasmid, pSEVA331 under an IPTG inducible promoter and both plasmids were transformed into a ΔbaeSΔcpxA double KO strain by heat shock transformation. Cultures were started by diluting overnights 200–500 fold into fresh LB medium +50 µg/mL AMP and allowed to grow to mid-log phase (OD$_{600}$ = 0.4–0.6) before analysis by flow cytometry. The responsiveness of cpxP reporter was confirmed by treating

log-phase cultures with 2 µg/mL brilacidin for 1.5 hr before analysis. Expression of HKs was confirmed by western analysis using the c-terminal 6 x His-tag for quantification.

## Flow cytometry

LB cultures at mid-log phase were diluted 20 x into 1 x PBS buffer and 20,000 cells gated by forward and side-scatter were evaluated for GFP fluorescence (p*cpxP*::GFP; Ex. 488 nm, Em. 515 nm) or mCherry fluorescence (p*spy*::mCherry, Ex. 488 nm, Em. 620 nm) per sample on a BD FACS caliber instrument. Sample average fluorescence and standard error were determined by standard analysis using Flo-Jo software.

## Data fitting

For data fitting, only data-sets in which kinase activity and sensor crosslinking have been determined simultaneously from the same samples at all 5 concentrations of $Mg^{2+}$ were included in analysis. The resulting kinase-active and sensor cross-linking-competent states are partitioned to generate expressions dependent on $[Mg^{2+}]$ as the lone variable as shown below. The parameters are then fit globally across all datasets, except for those accounting for the perturbation of a mutation/ $Gly_7$ disconnection, which are fit locally. Locally fit parameters are kept identical between replicates or additive mutations.

*Equation 1.* concerted model equation

$$F(ON) = \frac{PhoQ_{ON}+PhoQ_{ON}:Mg^{2+}}{PhoQ_{ON}+PhoQ_{ON}:Mg^{2+}+PhoQ_{OFF}+PhoQ_{OFF}:Mg^{2+}}$$
$$= \frac{\left(1+\frac{[Mg^{2+}]}{k_{dON}}\right)*K}{\left(1+\frac{[Mg^{2+}]}{k_{dON}}\right)^2*K+\left(1+\frac{[Mg^{2+}]}{K_{dOFF}}\right)^2} \tag{1}$$

*Equation 2.* two-domain two-state model fitting

$$F(Sensor_{ON}) = \frac{Sen_{ON}AK_{OFF}+Sen_{ON}AK_{ON}}{Sen_{ON}AK_{OFF}+Sen_{ON}AK_{ON}+Sen_{OFF}AK_{OFF}+Sen_{OFF}AK_{ON}}$$
$$= \frac{\left(1+\frac{[Mg^{2+}]}{k_{dON}}\right)*(k_{Sen}+\alpha K_{Sen}K_{AK})}{\left(1+\frac{[Mg^{2+}]}{k_{dON}}\right)*(K_{Sen}+\alpha K_{Sen}K_{AK})+\left(1+\frac{[Mg^{2+}]}{K_{dOFF}}\right)*(1+K_{AK})} \tag{2}$$

$$F(AutoKin_{ON}) = \frac{Sen_{ON}AK_{ON}+Sen_{OFF}AK_{ON}}{Sen_{ON}AK_{OFF}+Sen_{ON}AK_{ON}+Sen_{OFF}AK_{OFF}+Sen_{OFF}AK_{ON}}$$
$$= S*\frac{\left(1+\frac{[Mg^{2+}]}{k_{dON}}\right)^2\alpha K_{Sen}K_{AK}+\left(1+1+\frac{[Mg^{2+}]}{k_{dOFF}}\right)*K_{AK}}{\left(1+\frac{[Mg^{2+}]}{k_{dON}}\right)^2*(K_{Sen}+\alpha K_{Sen}K_{AK})+\left(1+\frac{[Mg^{2+}]}{k_{dOFF}}\right)^2 8(1+K_{AK})}$$

*Equation 3.* three-domain two-state model fitting

$$F(Sensor_{ON}) = \frac{Sen_{ON}HAMP_1AK_{OFF}+Sen_{ON}HAMP_1AK_{ON}+Sen_{ON}HAMP_2AK_{OFF}+Sen_{ON}HAMP_2AK_{ON}}{All}$$
$$= \frac{\left(1+\frac{[Mg^{2+}]}{K_{dON}}\right)*(K_{Sen}+k_{Sen}K_{AK}+\alpha_1 K_{Sen}K_{HAMP}+\alpha_1\alpha_2 K_{Sen}K_{HAMP}K_{AK})}{\left(1+\frac{[Mg^{2+}]}{K_{dON}}\right)*(K_{Sen}+k_{Sen}K_{AK}+\alpha_1 K_{Sen}K_{HAMP}+\alpha_1\alpha_2 K_{Sen}K_{HAMP}K_{AK})+\left(1+\frac{[Mg^{2+}]}{K_{dON}}\right)*(1+K_{AK}K_{HAMP}+\alpha_2)K_{HAMP}K_{AK}}$$

$$F(AutoKin_{ON}) = \frac{Sen_{ON}HAMP_1AK_{ON}+Sen_{ON}HAMP_2AK_{ON}+Sen_{ON}HAMP_1AK_{ON}+Sen_{ON}HAMP_2AK_{ON}}{All}$$
$$= \frac{\left(1+\frac{[Mg^{2+}]}{K_{dOFF}}\right)*(K_{AK}+\alpha_2 K_{HAMP}K_{AK})+\left(1+\frac{[Mg^{2+}]}{K_{dON}}\right)^2*(K_{Sen}K_{AK}+\alpha_1\alpha_2 K_{HAMP}K_{AK})}{\left(1+\frac{[Mg^{2+}]}{K_{dON}}\right)^2*(K_{Sen}+K_{Sen}K_{AK}+\alpha_1 K_{Sen}K_{HAMP}+\alpha_1\alpha_2 K_{Sen}K_{HAMP}K_{AK})+\left(1+\frac{[Mg^{2+}]}{K_{dOFF}}\right)*(1+K_{AK}K_{HAMP}+\alpha_2 K_{HAMP}K_{AK})}$$

To ensure equal weights in global fitting, the activity data was scaled by a factor of q = (mean of activity data) / (mean of %crosslink data). The crosslinking data and refactored activity data (Activity / q) were then globally fit to a three-state allosteric model. Each of 56 datasets (including replicates) was fit by a combination of global and local parameters, described in *Table 2*. Global parameters

**Table 2.** parameters used in fitting.

Values in red font indicate parameters fixed to one to account for $Gly_7$ insertion.

| Par. | Fit value | Lower bound | Upper bound | Fit datasets affected |
|---|---|---|---|---|
| $K_{dON}$ | 1.6 E-02 | 1.0 E-08 | 1.0 E + 02 | ALL |
| $K_{dOFF}$ | 3.7 E-04 | 1.0 E-08 | 1.0 E + 02 | ALL |
| $\alpha_1$ | 5.3 E-03 | 1.0 E-05 | 1.0 E + 05 | Y60C_Gly$_7$ 270/271, Y60C_Gly$_7$ 260/261, Y60C_Gly$_4$ 260/261, Y60C, Y40W, Y265A_Gly$_7$ 270/271, Y265A, V191W, S43W, R269L, R256A, R245F, R236A, N255A Gly7 260/261, N255A, N240A, L258A, L254A, E55S, E55A, E261F, E233A, E232A_Gly$_7$ 260/261, E232A |
| $\alpha_2$ | 1.0 E-08 | 1.0 E-08 | 1.0 E + 02 | Y60C_Gly$_7$ 219/220, Y60C, Y40W, V191W, S43W, S217_Gly$_7$ 219/220, S217W, R245F, R236A, N240A, L224F, L224A, L210A, I221F, I207A, E55S, E55A, A225F, A213W |
| $K_{Sen}$ | 9.5 E + 02 | 1.0 E-05 | 1.0 E + 05 | Y60C_Gly$_7$ 219/220, Y60C_Gly$_7$ 270/271, Y60C_Gly$_7$ 260/261, Y60C_Gly$_4$ 260/261, Y60C, Y265A_Gly$_7$ 270/271, Y265A, S217W_Gly$_7$ 219/220, S217W_Gly$_7$ 260/261, S217W, R269L, R256A, R245F, R236A, N255A_Gly$_7$ 260/261, N255A, N240A, L258A, L254A, L224F, L224A, L210A, I221F, I207A, E261F, E233A, E232A_Gly$_7$ 260/261, E232A |
| $K_{HAMP}$ | 2.2 E + 01 | 1.0 E-05 | 1.0 E + 05 | Y60C_Gly$_7$ 219/220, Y60C, Y40W, Y265A_Gly$_7$ 219/220, Y265A_Gly$_7$ 270/271, Y265A, V191W, S43W, R269L, E55S, E55A, A213W |
| $K_{AK}$ | 1.4 E-03 | 1.0 E-05 | 1.0 E + 05 | Y60C_Gly$_7$ 219/220, Y60C_Gly$_7$ 260/261, Y60C_Gly$_4$ 260/261, Y60C, Y40W, V191W, S43W, S217W_Gly$_7$ 219/220, S217W_Gly$_7$ 260/261, S217W, R256A, R245F, R236A, N255A_Gly$_7$ 260/261, N255A, N240A, L258A, L254A, L224F, L224A, L210A, I221F, I207A, E55S, E55A, E261F, E233A, E232A_Gly$_7$ 260/261, E232A, A225F, A213W |
| S | 7.4 E + 02 | 1.0 E-05 | 1.0 E + 05 | ALL |
| $K_{Sen}$ | 1.4 E + 03 | 1.0 E-05 | 1.0 E + 05 | Y40W |
| $K_{Sen}$ | 3.8 E + 02 | 1.0 E-05 | 1.0 E + 05 | S43W |
| $K_{Sen}$ | 4.1 E + 02 | 1.0 E-05 | 1.0 E + 05 | E55A |
| $K_{Sen}$ | 1.5 E + 03 | 1.0 E-05 | 1.0 E + 05 | E55S |
| $K_{Sen}$ | 1.2 E + 03 | 1.0 E-05 | 1.0 E + 05 | V191W |
| $K_{Sen}$ | 6.9 E + 02 | 1.0 E-05 | 1.0 E + 05 | I207A |
| $\alpha_1$ | 1.1 E-01 | 1.0 E-05 | 1.0 E + 05 | I207A |
| $K_{Sen}$ | 3.1 E-03 | 1.0 E-05 | 1.0 E + 05 | L210A |
| $\alpha_1$ | 9.1 E + 04 | 1.0 E-05 | 1.0 E + 05 | L210A |
| $\alpha_1$ | 1.0 E-05 | 1.0 E-05 | 1.0 E + 05 | A213W |
| $K_{Sen}$ | 4.0 E + 04 | 1.0 E-05 | 1.0 E + 05 | A213W |
| $K_{HAMP}$ | 7.1 E-01 | 1.0 E-05 | 1.0 E + 05 | S217W_Gly$_7$ 219/220, S217W_Gly$_7$ 260/261, S217W |
| $\alpha_1$ | 7.6 E-01 | 1.0 E-05 | 1.0 E + 05 | S217W_Gly$_7$ 260/261, S217W |
| $K_{HAMP}$ | 2.0 E + 01 | 1.0 E-05 | 1.0 E + 05 | I221F |
| $\alpha_1$ | 1.5 E-01 | 1.0 E-05 | 1.0 E + 05 | I221F |
| $K_{HAMP}$ | 1.6 E + 01 | 1.0 E-05 | 1.0 E + 05 | L224A |
| $\alpha_1$ | 1.3 E-01 | 1.0 E-05 | 1.0 E + 05 | L224A |
| Par. | Fit value | lower bound | Upper bound | Fit datasets affected |

*Table 2 continued on next page*

*Table 2 continued*

| Par. | Fit value | Lower bound | Upper bound | Fit datasets affected |
|---|---|---|---|---|
| $K_{HAMP}$ | 6.8 E + 01 | 1.0 E-05 | 1.0 E + 05 | L224F |
| $\alpha_1$ | 1.2 E-02 | 1.0 E-05 | 1.0 E + 05 | L224F |
| $K_{HAMP}$ | 2.4 E + 01 | 1.0 E-05 | 1.0 E + 05 | A 225F |
| $\alpha_1$ | 2.3 E-01 | 1.0 E-05 | 1.0 E + 05 | A 225F |
| $K_{Sen}$ | 1.0 E + 03 | 1.0 E-05 | 1.0 E + 05 | A 225F |
| $K_{HAMP}$ | 1.1 E + 02 | 1.0 E-05 | 1.0 E + 05 | E232A_Gly$_7$ 260/261, E232A |
| $\alpha_2$ | 1.4 E + 00 | 1.0 E-08 | 1.0 E + 02 | E232A |
| $K_{HAMP}$ | 2.4 E + 01 | 1.0 E-05 | 1.0 E + 05 | E233A |
| $\alpha_2$ | 1.0 E-08 | 1.0 E-08 | 1.0 E + 02 | E233A |
| $K_{HAMP}$ | 8.6 E + 00 | 1.0 E-05 | 1.0 E + 05 | R236A |
| $K_{HAMP}$ | 1.7 E + 01 | 1.0 E-05 | 1.0 E + 05 | N240A |
| $K_{HAMP}$ | 2.2 E + 01 | 1.0 E-05 | 1.0 E + 05 | R245F |
| $K_{AK}$ | 1.0 E-05 | 1.0 E-05 | 1.0 E + 05 | L254A |
| $\alpha_2$ | 1.0 E-08 | 1.0 E-08 | 1.0 E + 02 | L254A |
| $K_{AK}$ | 1.3 E-03 | 1.0 E-05 | 1.0 E + 05 | N255A_Gly$_7$ 260/261, N255A |
| $\alpha_2$ | 4.9 E-01 | 1.0 E-08 | 1.0 E + 02 | N255A |
| $K_{HAMP}$ | 3.6 E + 01 | 1.0 E-05 | 1.0 E + 05 | R256A |
| $\alpha_2$ | 1.0 E-08 | 1.0 E-08 | 1.0 E + 02 | R256A |
| $K_{AK}$ | 1.0 E-05 | 1.0 E-05 | 1.0 E + 05 | L258A |
| $\alpha_2$ | 1.0 E-08 | 1.0 E-08 | 1.0 E + 02 | L258A |
| $K_{HAMP}$ | 3.9 E + 01 | 1.0 E-05 | 1.0 E + 05 | E261F |
| $\alpha_2$ | 9.9 E-01 | 1.0 E-08 | 1.0 E + 02 | E261F |
| $K_{AK}$ | 4.1 E-04 | 1.0 E-05 | 1.0 E + 05 | Y265A_Gly$_7$ 219/220, Y265A_Gly$_7$ 270/271, Y265A |
| $\alpha_2$ | 3.2 E + 00 | 1.0 E-08 | 1.0 E + 02 | Y265A_Gly$_7$ 219/220, Y265A |
| $K_{AK}$ | 3.3 E-03 | 1.0 E-05 | 1.0 E + 05 | R269L |
| $\alpha_2$ | 1.0 E-08 | 1.0 E-08 | 1.0 E + 02 | R269L |
| $K_{HAMP}$ | 4.5 E + 01 | 1.0 E-05 | 1.0 E + 05 | Y60C_Gly$_7$ 260/261, Y60C_Gly$_4$ 260/261 |
| $\alpha_2$ | 1.0 E + 00 | | | Y60C_Gly$_7$ 260;261, Y60C_Gly$_4$ 260/261 |
| $\alpha_2$ | 1.0 E + 00 | | | S217W_Gly$_7$ 260/261 |
| $\alpha_2$ | 1.0 E + 00 | | | E232A_Gly$_7$ 260/261 |
| $\alpha_2$ | 1.0 E + 00 | | | N255A_Gly$_7$ 260/261 |
| $\alpha_1$ | 1.0 E + 00 | | | Y60C_Gly$_7$ 219/220 |
| $\alpha_1$ | 1.0 E + 00 | | | S217W_Gly$_7$ 219/220 |
| $\alpha_1$ | 1.0 E + 00 | | | Y265A_Gly$_7$ 219/220 |
| $K_{HAMP}$ | 2.1 E + 01 | 1.0 E-05 | 1.0 E + 05 | Y60C_Gly$_7$ 270/271 |
| $K_{AK}$ | 7.8 E-04 | 1.0 E-05 | 1.0 E + 05 | Y60C_Gly$_7$ 270/271 |
| $\alpha_2$ | 1.7 E + 00 | 1.0 E-08 | 1.0 E + 02 | Y60C_Gly$_7$ 270/271 |
| $\alpha_2$ | 3.8 E + 00 | 1.0 E-08 | 1.0 E + 02 | Y265A_Gly$_7$ 270/271 |

were shared between replicate datasets as well as datasets of mutations that were functionally similar. A total of 62 parameters (global and local, *Table 2*) were optimized using the python code found in the supplement (phoq_fit_local_global.py), from many rounds of fitting starting with random initial conditions (125,000 independent fits). Error analysis of the best-fit parameters (minimized sum of squares of residuals) was performed through bootstrapping of residuals with replacement to calculate confidence intervals, as well as residual sweep analyses (see below). To create synthetic bootstrapped datasets, we chose residuals at random with replacement and added these residuals to the activity and %crosslink values from the optimum fit. For each synthetic dataset, parameters were re-optimized, starting from initial values taken from the optimum fit. Out of 10,000 generated datasets 3061 fits were determined to have converged. The optimization process was considered converged when the cost function F did not change considerably (dF< ftol * F, with ftol = 1e-8, that is, convergence criterion two from Scipy least_squares). Histograms of these bootstrapped parameter values show the spread in possible values due to errors in the fit (*Figure 7D* and *Figure 8—figure supplement 1*). Analysis of the bootstrapped parameter distributions showed correlations between the globally fit parameters S and $K_{AK}$ (*Figure 7—figure supplement 2*).

We also performed a residual sweep analysis to assess the quality of the fit in response to changes in a single parameter value, with all other parameters held fixed. For residual sweep analysis, all but one of the parameters were fixed to their optimum values, and the variable under analysis was swept across its allowed numerical range, after which the sum of squares of residuals was calculated. The sum of squares was then plotted as a function of the parameter's numerical value (*Figure 7D* and *Figure 8—figure supplement 1*). Code to reproduce the fits and plots is given in the comment section at the bottom of the supplement python scripts (phoq_fit_local_global.py, phoq_fit_local_global_ipython.py, and phoq_fit_ci_local_global.py). Scripts to run the fitting on the UCSF Wynton High Performance Computing cluster can also be found in the supplement (phoq_fit.job and phoq_fit_ci.job).

## Choice of locally varied parameters

Mutations contained entirely within a given domain are allowed to vary the intrinsic equilibrium of that domain only. Mutations within 1 heptad of a domain-domain junction (219/220 for sensor/HAMP, 260/261 for HAMP/autokinase) are also allowed to vary the equilibrium constant of the domain they reside in, as well as the coupling constant between the two domains. Exceptions to this rule include A225F, which was additionally allowed to vary the $K_{Sen}$ parameter, along with $K_{HAMP}$ and $\alpha_1$ parameters which would normally be varied. Given the poor fit to this mutant, we hypothesized that the disruption of inserting a large Phe sidechain in place of an alanine may propagate into the preceding transmembrane region. Similarly, we allowed $K_{AK}$ to float locally for L254A, N255A, and L258A, which resulted in better fits as discussed in main text. Finally, $\alpha_2$ was allowed to float for E231A and E232A, which have been hypothesized in previous work to directly couple to the autokinase domain via a salt-bridge to an arginine residue in the autokinase (*Matamouros et al., 2015*).

## Integrative modeling and Molecular dynamics

Rosetta (*Leaver-Fay et al., 2011*), a powerful protein design suite, was employed to produce the initial model of PhoQ. The integrative modeling procedure was used to combine X-ray structures of the PhoQ sensor PDB id: 3bq8 (*Cheung et al., 2008*) and CpxA kinase domain PDB id: 4biv, *Mechaly et al., 2014*, with the atomic model of PhoQ transmembrane domain (*Lemmin et al., 2013*).

A Molecular Dynamics (MD) simulation was carried out to further refine the PhoQ model. The structure was embedded into a phosphatidylcholine (POPC) membrane, solvated in a 17 Å padding water box, and neutralized by the addition of NaCl salt at a concentration of 150 mM. No ligands ($Mg^{2+}$, nucleotides) were present in the simulation. The simulation was performed with the NAMD MD engine (*Phillips et al., 2020*) and the CHARMM36 force field (*Huang and MacKerell, 2013*). TIP3P water parameterization was used to describe the water molecules. The periodic electrostatic interactions were computed using particle-mesh Ewald (PME) summation and a grid spacing smaller than 1 Å. Constant temperature of 310 K was imposed with Langevin dynamics, and constant pressure of 1 atm was maintained with a Langevin barostat. During equilibration, the position of the backbone atoms was restrained with harmonic restraints. The system was minimized by 5000 conjugate gradient

steps and followed by a 20 ns equilibration. The positional restraints were then replaced with the secondary structure restraints. The molecular dynamics simulation was performed up to 100 ns.

## Acknowledgements

We thank Dr. Mark Goulian for many helpful discussions and sharing *E coli* reporter strains.

## Additional information

### Competing interests

Kathleen S Molnar: is an employee of Codexis Inc. The author declares that no other competing interests exist. The other authors declare that no competing interests exist.

### Funding

| Funder | Grant reference number | Author |
|---|---|---|
| National Institutes of Health | K99-GM138753 | Bruk Mensa<br>Nicholas F Polizzi<br>Andrew M Natale<br>Thomas Lemmin<br>William F DeGrado |

The funders had no role in study design, data collection and interpretation, or the decision to submit the work for publication.

### Author contributions

Bruk Mensa, Conceptualization, Data curation, Formal analysis, Investigation, Methodology, Visualization, Writing – original draft, Writing – review and editing; Nicholas F Polizzi, Formal analysis, Methodology, Software, Visualization, Writing – original draft, Writing – review and editing; Kathleen S Molnar, Conceptualization, Data curation, Formal analysis, Methodology, Writing – review and editing; Andrew M Natale, Data curation, Methodology, Writing – review and editing; Thomas Lemmin, Methodology, Visualization, Writing – review and editing; William F DeGrado, Conceptualization, Formal analysis, Funding acquisition, Investigation, Methodology, Supervision, Writing – original draft, Writing – review and editing

### Author ORCIDs

Bruk Mensa http://orcid.org/0000-0002-8777-5946

### Decision letter and Author response

Decision letter https://doi.org/10.7554/eLife.73336.sa1
Author response https://doi.org/10.7554/eLife.73336.sa2

## Additional files

### Supplementary files

- Transparent reporting form

- Source code 1. HPC job submission file to run global fitting of PhoQ activity/crosslinking data on UCSF's wynton compute cluster.

- Source code 2. HPC job submission file to run bootstrap fitting for confidence interval analysis of the global PhoQ fit.

- Source code 3. Python script slightly changed from 'phoq_fit_local_global.py' to load properly in an iPython session.

- Source code 4. Python script containing code for global and local fitting of multi-state models to PhoQ activity/crosslinking data.

- Source code 5. Python script containing code fto run bootstrap fitting for confidence intervals of the global PhoQ fit.

• Source code 6. Three-domain, two-state model parameter fit values for PhoQ mutant dataset.

## Data availability

All data generated or analysed during this study are included in the manuscript and supporting file; Source data files have been provided for Figure 8. All source code for modeling work is provided as source code files 1-5.

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
