## [Editor Report]

This paper examines the mechanism of signal transmission through the histidine kinase PhoQ. The paper nicely describes a model of signaling by allosteric coupling between individual domains rather than by a concerted conformational change and provides substantial experimental evidence for the model from characterization of over 30 mutational substitutions. The allosteric coupling model provides a way to understand many diverse observations about signaling by two-component receptors and has the potential to be relevant to conformational signaling by many other transmembrane receptors.

---

## [Decision Letter]

**Decision letter after peer review:**

Thank you for submitting your article "Allosteric mechanism of signal transduction in the two-component system histidine kinase PhoQ" for consideration by *eLife*. Your article has been reviewed by 3 peer reviewers, and the evaluation has been overseen by a Reviewing Editor and Volker Dötsch as the Senior Editor. The reviewers have opted to remain anonymous.

Essential revisions:

This paper, and the proposed model of transmembrane signaling mediated by allosteric coupling, was enthusiastically received by the reviewers and editor. Although generally positive, the reviewers raised a number of concerns (see below) that should be addressed in a revision. In particular, the authors should:

1) Revise the writing substantially to make the paper more accessible to non-enzymologists. The last paragraph of Reviewer #2's public review provides a useful roadmap to a more accessible manuscript.

2) Revise their model based on the feedback of Reviewer #3 about how Mg binding is handled.

*Reviewer #1 (Recommendations for the authors):*

The model in which is suggested that PhoQ transmit signals via interdomain allostery is statistically justified? The other possible models do not fit statistically?

It is not clear to me why in Figure 2A and 7A is necessary a MD model to indicate the different domains of the protein. It seems that will be sufficient a sketch of the different domains of the protein.

*Reviewer #2 (Recommendations for the authors):*

1. How were the positions of the mutational substitutions chosen and how are they related to the protein structure? Except for the introduction of Trp at membrane boundaries and noting for other substitutions their general positions in the three-dimensional structure in Figure 7A and that they were "expected to be on the interior of the protein", no information is provided. More information would help an interested reader understand the logic of mutational placement.

2. Why were so few substitutions placed in the sensor domain since there were many possibilities away from the Mg2+-binding site? It seems as if they would have been informative.

3. It is not clear why the authors avoid mutational substitutions in the entire autokinase domain, since there are many positions removed from the active site. More of an explanation should be provided I the text or Materials and methods.

4. What about the role of the transmembrane domain? In the manuscript, this region is considered part of the sensor domain. What is the basis for this consideration? Could it be a distinct domain with its own equilibrium between on and off? This issue should be discussed in the text.

5. The mutant Gly7 270/271 is shown in Figure 7A, listed in Table 2 alone and in combination with single-site substitutions, and cited on line 522 but is not otherwise discussed in the text. It should be.

*Reviewer #3 (Recommendations for the authors):*

This manuscript describes a comprehensive study of kinase activation and allosteric coupling in the sensor histidine kinase (SHKs) PhoQ. Quantitative assays for sensor domain activation and kinase response are used to evaluate a large number of variant proteins that display a range of properties with respect to ligand binding, interdomain coupling and kinase activity. The data is used to construct and fit a conceptually elegant model that provides a thermodynamic explanation for domain interactions, allostery and sensing responses in SHKs. The experiments also demonstrate that sensor kinase domains intrinsically favor their "on" states and that HAMP domains act to deactivate both the sensor and the kinase units. In all it is a very impressive study that sets the bar for enzymatic approaches aimed at understanding signaling by multidomain transmembrane kinases. Generality of key principles are explored by examining several SHKs related to PhoQ. The paper is well written and the complex data and their interpretation are for the most part clearly discussed. That said, there are some issues the authors should address:

The model applied for Mg binding should be described to a greater extent. The equations of Figures 1, 2,3,6,7 represent a situation more complex than the accompanying schematics portray. Even the simplest equation of 1B implies sequential binding of 2 Mg ions to one PhoQ dimer (presumably 1 site per subunit). Furthermore, the binding sites are assumed to be independent and, importantly, there are no intermediate states in the model in which one subunit is "on" (in either its sensor or kinase domain) and the other is "off". Is it known experimentally that the two subunits act independently and what is the consequence of not allowing for hybrid activation states within the dimer?

In addition, there may be a factor of 2 missing in treating the relative dissociation constants for Mg binding to an empty PhoQ or to a singly Mg-occupied PhoQ. Because the multiplicity changes by a factor of 2 in going from both the empty to the half-occupied state and again by 2 in going from the half-occupied to the fully occupied states, the effective Kd for binding to the singly occupied state is 4x larger than for binding to the empty state. It appears that all of the models accommodate only a factor of 2. This issue affects the (1 + [Mg]/Kd)2 term, likely to a minor extent.

In a similar vein, for the final models of Figure 6,7, why is Mg binding only considered to selected states (SenOFF/HAMP1/Akon/off, for example)? And in Figure 6A what does AK "on/off" signify?

Line 549 – Discussion of the setpoint of the autokinase domain depends on the "reference point" given that KAK and alpha2 are correlated parameters. For example, one could view the intrinsic activity of the autokinase as being the fully uncoupled state, with KAK defined closed to 1.0 and alpha2 having a smaller value that currently modeled in the case of the Y60C (WT) protein. Could one fix KSen and KAK at the values for the Gly-decoupled systems and allow the shifts in equilibrium owing to HAMP coupling to be compensated solely for by alpha1 and alpha2? This framing might be more straightforward for understanding the HAMP coupling. Although the reference position is largely arbitrary and in any given fitting scheme likely depends on the choice of constraining and fixing parameters, it does alter how one views the role of kinase-activating mutations. i.e. with the fully decoupled state as the reference, the HAMP is always deactivating, with different variants (including the WT) deactivated to varying extents. Some additional comments on this issue may help readers understand the range of kinase behavior and how it is influenced by HAMP.

Related to the previous point, in Figure 7 the alpha2 parameter seems to have a large amount of uncertainty, and appears biphasic in the fits, this behavior deserves a comment as to its impact in the model. How much would the interpretations change if alpha2 is considered to hold its extreme values?

p. 30 line 587 – It's unclear what is meant by the statement that the HAMP domain "serves to tune the ligand-sensitivity amplitude of the response" (p. 30 line 587). In this model, the HAMP domain does alter the sensitivity of the sensor domain by favoring the sensor OFF state (even though it does not directly modulate KdOFF), but what is meant by "sensitivity amplitude".

Has the MD model referenced in Figure 2 been published previously? If not, some information on its production should be provided.

Representative data for the cross-linking and western blotting should be shown in the supplemental.

[Editors’ note: further revisions were suggested prior to acceptance, as described below.]

Thank you for resubmitting your work entitled "Allosteric mechanism of signal transduction in the two-component system histidine kinase PhoQ" for further consideration by *eLife*. Your revised article has been evaluated by Volker Dötsch (Senior Editor) and a Reviewing Editor.

The manuscript has been improved but there are some remaining issues that need to be addressed – see the comments from Reviewer 2 and 3 below.

*Reviewer #1 (Recommendations for the authors):*

The authors answered satisfactorily my commentaries and concerns. The way in which the authors wrote the revised manuscript is much more ordered and comprehensive. This work represents an interesting attempt to explain the mechanism of signal transduction by sensor kinases in two-component systems.

*Reviewer #2 (Recommendations for the authors):*

The authors have in large part addressed the concerns I had expressed about the initial version of this submission. Most importantly, these include a major reorganization of the manuscript by presenting experimental data before discussing models that could explain the data. This reorganization has resulted in an improved manuscript. However, the reorganized manuscript includes text that is now unnecessary, given the new order of presentation. Eliminating such text would shorten the paper and make it easier for an interested reader to follow the essential line of reasoning. The authors should critically edit a clean copy of the revised text for brevity with the aim of eliminating unnecessary repetition. For instance, there is no need for the text in lines 127-143 since similar statements are made in the Introduction, lines 121-125 or in the section that begins on line 183. A second example is lines 570-584 where there is no need to describe again the mutations that have already been explained in previous sections (although mention of Gly7 260/261 need to be included in the earlier section about Gly7 insertions).

*Reviewer #3 (Recommendations for the authors):*

The authors have done a good job of responding to my comments. The new discussion in lines 254-267 regarding the Mg binding (Lines 423-438 in the edited version) is very helpful. I do, however, still think that it should be explicitly stated somewhere that there are no hybrid subunit states (one subunit on and one subunit off) considered in the model. The data may indeed not distinguish fully cooperative from non-cooperative behavior in terms of subunit activation, but the readers should realize that the subunits are considered independent for Mg binding but not so for activation.

---

## [Author Response]

Essential revisions:This paper, and the proposed model of transmembrane signaling mediated by allosteric coupling, was enthusiastically received by the reviewers and editor. Although generally positive, the reviewers raised a number of concerns (see below) that should be addressed in a revision. In particular, the authors should:1) Revise the writing substantially to make the paper more accessible to non-enzymologists. The last paragraph of Reviewer #2's public review provides a useful roadmap to a more accessible manuscript.

Per Reviewer #2’s suggestion, we have made the following changes to bring forward and emphasize the experimentally generated results and consolidate modeling work in the latter half of the manuscript. We have also added several clarifications to make text more accessible to non-enzymologists.

1. New Figure 2 now contains an expanded set of experimental data (Figure 2B, 2C) to provide motivation for the subsequent mathematical analysis.

2. New Figure 2—figure supplement 1 provides examples of western blots and crosslinking quantifications.

3. We have moved the former Figure 4 on Gly_7_ insertions in PhoQ forward to Figure 3 along with the associated Results section.

4. We have moved the former Figure 5 on Gly_7_ insertions in other HKs forward to Figure 4 along with the associated Results section.

5. We have moved Figure 7A into figure 7—figure supplement 1 and expanded on the justification of mutations and Gly_7_ insertion choices.

6. We have consolidated the concerted (1-domain) signaling model from Figure 2 (formerly Figure 2B, 2D) and the 2-domain allosteric model from the former Figure 3 into a new figure, Figure 5. Certain elements of this figure have been migrated to Figure 5—figure supplement 1 to aid in visual clarity. The Results sections for these parts have been consolidated to lines 250-355

2) Revise their model based on the feedback of Reviewer #3 about how Mg binding is handled.

We have elaborated on how Mg^2+^ binding is handled in the model (Lines 254-267). We discuss the factors that are built into the “observed” K_d_ that is reported in the manuscript, as a result of the formulation of our model.

Reviewer #1 (Recommendations for the authors):The model in which is suggested that PhoQ transmit signals via interdomain allostery is statistically justified? The other possible models do not fit statistically?

We have added Figure 7—figure supplement 4 showing poor fitting results for the simpler 2-domain model. We have made additional comments in Results (lines 474-478, 513-519) and Methods (lines 831-832) on this subject, and supplied the full list of fit values for the 2-domain 2-state model as “Figure 7—figure supplement 4 Source data file 1”.

It is not clear to me why in Figure 2A and 7A is necessary a MD model to indicate the different domains of the protein. It seems that will be sufficient a sketch of the different domains of the protein.

We have now place nine selected mutants in diagrammatic form in Figure 2B, as suggested by the reviewer, but it became too busy when all were included. For the full set of mutants we felt it useful to present mutation locations on an MD model particularly to show buried vs. solvent accessible positions. Furthermore, this representation better illustrates the tertiary context of mutations at the HAMP/autokinase junction. We have migrated Figure 7A to Figure 7—figure supplement 1 and highlighted the justification for our choice in mutations and Gly_7_ insertions using this MD model. We have provided a paragraph in the Methods section detailing the generation of the model (lines 878-894).

Reviewer #2 (Recommendations for the authors):1. How were the positions of the mutational substitutions chosen and how are they related to the protein structure? Except for the introduction of Trp at membrane boundaries and noting for other substitutions their general positions in the three-dimensional structure in Figure 7A and that they were "expected to be on the interior of the protein", no information is provided. More information would help an interested reader understand the logic of mutational placement.

We have moved Figure 7A to Figure 7—figure supplement 1 and expanded on it to include rationale for choice of mutations and Gly_7_ insertions. The choice of single point mutations is further discussed in lines 136-146, 189-194, 395-407. We also provide additional references (Refs 44-54).

2. Why were so few substitutions placed in the sensor domain since there were many possibilities away from the Mg2+-binding site? It seems as if they would have been informative.

In this work, we primarily try to understand how changes in the sensor domain are transduced across the HAMP domain to alter the activity of the autokinase domain. Ligand (Mg^2+^) binding remains the most robust way of altering the sensor domain equilibrium to assess its coupling to the rest of PhoQ. We incorporate some sensor mutants near the dimeric interface to show how the sensor’s intrinsic equilibrium can be altered by mutation; however, more sensor mutations would not add information content to examining the allosteric coupling to HAMP and autokinase beyond what is achieved by ligand binding. This rationale is discussed in lines 136-139.

3. It is not clear why the authors avoid mutational substitutions in the entire autokinase domain, since there are many positions removed from the active site. More of an explanation should be provided I the text or Materials and methods.

Similar to comment #2, we avoid mutations in the autokinase because we are interested in the modulation of this domains activity through coupling to upstream domains, rather than changes within the domain itself. Furthermore, the effects of mutations in the autokinase are difficult to interpret due to their likely involvement in one or more crucial processes for activity: nucleotide binding/release, formation of asymmetric catalytic states in PhoQ, specific recruitment of cognate response regulator and interactions involved in dephosphorylation. We do make mutations at the HAMP-autokinase junction that nonetheless seem to affect the basal activity of the autokinase (eg. L254A, N255A, L258A, Y265A) as discussed in Results (lines 593-599) and Methods (lines 874-875).

4. What about the role of the transmembrane domain? In the manuscript, this region is considered part of the sensor domain. What is the basis for this consideration? Could it be a distinct domain with its own equilibrium between on and off? This issue should be discussed in the text.

We note that our allosteric coupling model can be extended to include more modular domains (e.g., TM 4-helix bundle and S-helix). However, such granularity rapidly inflates the number of parameters needed to describe the corresponding model (number of states = # of domains ^ # of states). Given the already established structural coupling of ligand-binding to TM geometry in phoQ (reported in Molnar et al., 2014 (Ref 39)), we opted to treat it as part of the same ligand-sensing module (i.e. ‘sensor’) as discussed in Results (lines 560-563)

5. The mutant Gly7 270/271 is shown in Figure 7A, listed in Table 2 alone and in combination with single-site substitutions, and cited on line 522 but is not otherwise discussed in the text. It should be.

The Gly_7_ 270/271 mutant is now discussed in Results (lines 399-403).

Reviewer #3 (Recommendations for the authors):This manuscript describes a comprehensive study of kinase activation and allosteric coupling in the sensor histidine kinase (SHKs) PhoQ. Quantitative assays for sensor domain activation and kinase response are used to evaluate a large number of variant proteins that display a range of properties with respect to ligand binding, interdomain coupling and kinase activity. The data is used to construct and fit a conceptually elegant model that provides a thermodynamic explanation for domain interactions, allostery and sensing responses in SHKs. The experiments also demonstrate that sensor kinase domains intrinsically favor their "on" states and that HAMP domains act to deactivate both the sensor and the kinase units. In all it is a very impressive study that sets the bar for enzymatic approaches aimed at understanding signaling by multidomain transmembrane kinases. Generality of key principles are explored by examining several SHKs related to PhoQ. The paper is well written and the complex data and their interpretation are for the most part clearly discussed. That said, there are some issues the authors should address:The model applied for Mg binding should be described to a greater extent. The equations of Figures 1, 2,3,6,7 represent a situation more complex than the accompanying schematics portray. Even the simplest equation of 1B implies sequential binding of 2 Mg ions to one PhoQ dimer (presumably 1 site per subunit). Furthermore, the binding sites are assumed to be independent and, importantly, there are no intermediate states in the model in which one subunit is "on" (in either its sensor or kinase domain) and the other is "off". Is it known experimentally that the two subunits act independently and what is the consequence of not allowing for hybrid activation states within the dimer?

Addressed in response to public review #3

In addition, there may be a factor of 2 missing in treating the relative dissociation constants for Mg binding to an empty PhoQ or to a singly Mg-occupied PhoQ. Because the multiplicity changes by a factor of 2 in going from both the empty to the half-occupied state and again by 2 in going from the half-occupied to the fully occupied states, the effective Kd for binding to the singly occupied state is 4x larger than for binding to the empty state. It appears that all of the models accommodate only a factor of 2. This issue affects the (1 + [Mg]/Kd)2 term, likely to a minor extent.

Addressed in response to public review #3

In a similar vein, for the final models of Figure 6,7, why is Mg binding only considered to selected states (SenOFF/HAMP1/Akon/off, for example)? And in Figure 6A what does AK"on/off" signify?

Addressed in response to public review #3

Line 549 – Discussion of the setpoint of the autokinase domain depends on the "reference point" given that KAK and alpha2 are correlated parameters. For example, one could view the intrinsic activity of the autokinase as being the fully uncoupled state, with KAK defined closed to 1.0 and alpha2 having a smaller value that currently modeled in the case of the Y60C (WT) protein. Could one fix KSen and KAK at the values for the Gly-decoupled systems and allow the shifts in equilibrium owing to HAMP coupling to be compensated solely for by alpha1 and alpha2? This framing might be more straightforward for understanding the HAMP coupling. Although the reference position is largely arbitrary and in any given fitting scheme likely depends on the choice of constraining and fixing parameters, it does alter how one views the role of kinase-activating mutations. i.e. with the fully decoupled state as the reference, the HAMP is always deactivating, with different variants (including the WT) deactivated to varying extents. Some additional comments on this issue may help readers understand the range of kinase behavior and how it is influenced by HAMP.

Addressed in response to public review #3

Related to the previous point, in Figure 7 the alpha2 parameter seems to have a large amount of uncertainty, and appears biphasic in the fits, this behavior deserves a comment as to its impact in the model. How much would the interpretations change if alpha2 is considered to hold its extreme values?

Addressed in response to public review #3

p. 30 line 587 – It's unclear what is meant by the statement that the HAMP domain "serves to tune the ligand-sensitivity amplitude of the response" (p. 30 line 587). In this model, the HAMP domain does alter the sensitivity of the sensor domain by favoring the sensor OFF state (even though it does not directly modulate KdOFF), but what is meant by "sensitivity amplitude".

Addressed in response to public review #3

Has the MD model referenced in Figure 2 been published previously? If not, some information on its production should be provided.

A paragraph has been added to Methods section detailing generation of MD model (lines 878-894).

Representative data for the cross-linking and western blotting should be shown in the supplemental.

Figure 2—figure supplement 1 has been added with representative western blot and crosslinking quantifications.

[Editors’ note: further revisions were suggested prior to acceptance, as described below.]

Reviewer #2 (Recommendations for the authors):The authors have in large part addressed the concerns I had expressed about the initial version of this submission. Most importantly, these include a major reorganization of the manuscript by presenting experimental data before discussing models that could explain the data. This reorganization has resulted in an improved manuscript. However, the reorganized manuscript includes text that is now unnecessary, given the new order of presentation. Eliminating such text would shorten the paper and make it easier for an interested reader to follow the essential line of reasoning. The authors should critically edit a clean copy of the revised text for brevity with the aim of eliminating unnecessary repetition. For instance, there is no need for the text in lines 127-143 since similar statements are made in the Introduction, lines 121-125 or in the section that begins on line 183.

We have truncated the repeated statements in lines 127-143 of Results.

A second example is lines 570-584 where there is no need to describe again the mutations that have already been explained in previous sections (although mention of Gly7 260/261 need to be included in the earlier section about Gly7 insertions).

We have removed redundant descriptions of mutations. Gly_7_ 260/261 is first introduced on line 184.

Reviewer #3 (Recommendations for the authors):The authors have done a good job of responding to my comments. The new discussion in lines 254-267 regarding the Mg binding (Lines 423-438 in the edited version) is very helpful. I do, however, still think that it should be explicitly stated somewhere that there are no hybrid subunit states (one subunit on and one subunit off) considered in the model. The data may indeed not distinguish fully cooperative from non-cooperative behavior in terms of subunit activation, but the readers should realize that the subunits are considered independent for Mg binding but not so for activation.

We have altered lines 255-258 to state this explicitly.